# Offline Reinforcement Learning from Randomly Perturbed Data Sources

## Abstract

Most of the existing offline reinforcement learning (RL) studies assume the available dataset is sampled directly from the target task. However, in some practical applications, the available data are often coming from several related but heterogeneous environments. A theoretical understanding of efficient learning from heterogeneous offline datasets remains lacking. In this work, we study the problem of offline RL based on multiple data sources that are randomly perturbed versions of the target Markov decision process (MDP). A novel HetPEVI algorithm is first proposed, which simultaneously considers two types of uncertainties: sample uncertainties from a finite number of data samples per data source, and source uncertainties due to a finite number of data sources. In particular, the sample uncertainties from all data sources are jointly aggregated, while an additional penalty term is specially constructed to compensate for the source uncertainties. Theoretical analysis demonstrates the near-optimal performance of HetPEVI. More importantly, the costs and benefits of learning with randomly perturbed data sources are explicitly characterized: on one hand, an *unavoidable* performance loss occurs due to the indirect access to the target MDP; on the other hand, efficient learning is achievable as long as the sources *collectively* (instead of individually) provides a good data coverage. Finally, we extend the study to linear function approximation and propose the HetPEVI-Lin algorithm that provides additional efficiency guarantees beyond the tabular cases.

## 1 Introduction

Offline reinforcement learning (RL) (Levine et al., 2020), a.k.a. batch RL (Lange et al., 2012), has received growing interest in the recent years. It aims at training RL agents using accessible datasets collected *a priori* and thus avoids expensive online interactions. Along with its tremendous empirical successes (Kidambi et al., 2020), recent studies have also established theoretical understandings of offline RL (Rashidinejad et al., 2021; Jin et al., 2021b; Duan et al., 2021; Uehara & Sun, 2021).

Despite these advances, the majority of offline RL research focuses on learning via data collected exactly from the target task environment (Kumar et al., 2020). However, in practice, it is difficult to ensure that all such data are perfectly from one source environment. Instead, in many cases, it is more reasonable to assume that data are collected from different sources that are perturbed versions of the target task. For example, when training a chatbot (Jaques et al., 2020), the offline dialogue datasets typically consist of short conversations between different people, who naturally have varying language habits. The training objective is the common underlying language structure, e.g., basic grammar, which however cannot be reflected in any individual dialogue but must be holistically learned from the aggregation of them. More examples can be found in healthcare (Tang & Wiens, 2021), autonomous driving (Sallab et al., 2017) and others. While a few empirical investigations under the offline meta-RL framework have been reported (Dorfman et al., 2021; Lin et al., 2022; Mitchell et al., 2021), theoretical understandings of effectively and efficiently learning the underlying task using datasets from multiple heterogeneous sources are largely lacking.

Motivated by both practical and theoretical limitations, this work makes progress in the under-explored RL problem of *learning the target task using data from heterogeneously perturbed data sources*. In particular, we study the problem of learning a target Markov decision process (MDP)

from offline datasets sampled from multiple heterogeneously realized source MDPs. Several provably efficient designs are proposed, targeting both tabular and linear MDPs. To the best of our knowledge, this is the first work that proposes provably efficient offline RL algorithms to handle perturbed data sources, which can benefit relevant applications and further shed light on the theoretical understanding of offline meta-RL. The contributions are summarized as follows:

• We study a new offline RL problem where the datasets are collected from multiple heterogeneous source MDPs, with possibly different reward and transition dynamics, instead of directly from the target MDP. Motivated by practical applications, the data source MDPs are modeled as random perturbations of the target MDP. Compared with studies of offline RL using data directly from the target MDP (Rashidinejad et al., 2021; Jin et al., 2021b), we face new challenges of jointly considering uncertainties caused by the finite number of data samples per source (referred to as the *sample uncertainties*) and by the finite number of data sources (referred to as the *source uncertainties*).

• A novel HetPEVI algorithm is proposed, which generalizes the idea of pessimistic value iteration (Jin et al., 2021b) and uses carefully crafted penalty terms to address the sample and source uncertainties simultaneously. Specifically, in the HetPEVI, the specially designed penalty term contains two parts: one aggregating the sample uncertainties associated with each dataset, and the other term compensating the source uncertainties. The combination of these two parts jointly characterizes the uncertainty associated with the collected datasets.

• Theoretical analysis proves the effectiveness of HetPEVI with a corresponding lower bound, which first demonstrates that even with finite randomly perturbed MDPs and finite data samples from each of them, it is feasible to efficiently learn the target MDP. More importantly, the analysis reveals that learning with multiple perturbed data sources brings both costs and benefits. On one hand, due to indirect access to the target MDP, an unavoidable learning cost occurs. This cost scales only with the number of data sources and cannot be reduced by increasing the size of datasets, which highlights the importance of the diversity of data sources. On the other hand, effective learning only requires that the datasets collectively (instead of individually) provide a good coverage of the optimal policy, which may provide additional insights for practical data collections.

• Moreover, we extend the study to linear function approximation where offline data are collected from linear MDPs with a shared feature mapping but heterogeneously realized system dynamics. The HetPEVI-Lin algorithm is developed to jointly consider the sample and source uncertainties while incorporating the linear structure. Theoretical analysis demonstrates the effectiveness of HetPEVI-Lin and verifies the sufficiency of a good collective coverage.

**Related Works.** With the empirical success of offline RL (Levine et al., 2020), its theoretical understandings have been gradually established in recent years. In particular, the principle of "pessimism" is incorporated and proved efficient for offline RL (Jin et al., 2021b; Rashidinejad et al., 2021). Following this line, Xie et al. (2021b); Li et al. (2022); Shi et al. (2022) further fine-tune the designs for the tabular setting and Zanette et al. (2021); Min et al. (2021); Yin et al. (2022); Xiong et al. (2022) for linear MDPs (Jin et al., 2020). These theoretical advances are mainly focused on learning with data directly from the target task. However, in the practical studies of RL, growing interests have been made to utilize data from heterogeneous sources, e.g., offline meta-RL (Mitchell et al., 2021; Dorfman et al., 2021; Lin et al., 2022; Li et al., 2020b). This work is thus motivated to provide a theoretical understanding of how to extract information about the target task from multiple sources. A more detailed literature review regarding both online and offline RL with single or heterogeneous environments is provided in Appendix A.1.

## 2 PROBLEM FORMULATION

**Preliminaries of RL.** We consider an RL problem characterized by an episodic MDP $\mathcal{M} := (H, \mathcal{S}, \mathcal{A}, \mathbb{P}, r)$. In this tuple, $H$ is the length of each episode, $\mathcal{S}$ is the state space, $\mathcal{A}$ is the action space, $\mathbb{P}$ is the transition matrix so that $\mathbb{P}_h(s'|s, a)$ gives the probability of transiting to state $s'$ if action $a$ is taken for state $s$ at step $h$, and $r_h(s, a)$ is the deterministic reward in the interval of $[0, 1]$ of taking action $a$ for state $s$ at step $h$.[1] Specifically, in each episode, starting from an initial state $s_1$, at each step $h \in [H]$, the agent observes state $s_h \in \mathcal{S}$, picks action $a_h \in \mathcal{A}$, receives reward

---

[1]The assumption of deterministic rewards is standard in theoretical analysis of RL (Jin et al., 2018; 2020) as the uncertainties in estimating rewards are dominated by those in estimating transitions.

$r_h(s_h, a_h)$, and then transits to a next state $s_{h+1} \sim \mathbb{P}_h(\cdot|s_h, a_h)$. The episode ends after $H$ steps. In the tabular RL setting, the state and action spaces are finite with $S := |\mathcal{S}|$ and $A := |\mathcal{A}|$.

A policy $\pi := \{\pi_h(\cdot|s) : (s, h) \in \mathcal{S} \times [H]\}$ consists of distributions $\pi_h(\cdot|s)$ over the action space $\mathcal{A}$, whose value functions can be defined as $V_h^\pi(s) := \mathbb{E}_{\pi, \mathcal{M}}[\sum_{h'=h}^H r_{h'}(s_{h'}, a_{h'})|s_h = s]$ and $Q_h^\pi(s, a) := \mathbb{E}_{\pi, \mathcal{M}}[\sum_{h'=h}^H r_{h'}(s_{h'}, a_{h'})|s_h = s, a_h = a]$ for each $(s, a, h) \in \mathcal{S} \times \mathcal{A} \times [H]$, where the expectation $\mathbb{E}_{\pi, \mathcal{M}}[\cdot]$ is with respect to (w.r.t.) the random trajectory induced by policy $\pi$ on MDP $\mathcal{M}$. The optimal policy $\pi^*$ maximizes the value function, i.e., $\pi^* := \arg\max_\pi V_1^\pi(s)$. For convenience, we denote $V_h^*(s) := V_h^{\pi^*}(s)$ and $Q_h^*(s, a) := Q_h^{\pi^*}(s, a)$ for all $(s, a, h) \in \mathcal{S} \times \mathcal{A} \times [H]$, and use $\pi_h(s)$ to refer to the chosen action for state $s$ at step $h$ for a deterministic policy $\pi$.

## 2.1 THE LEARNING TARGET AND OFFLINE DATASETS

**The Target Task.** This work considers a target task modeled by an MDP $\mathcal{M} = (H, \mathcal{S}, \mathcal{A}, \mathbb{P}, r)$, which can be any task of interest (e.g., chatbot training in Sec. 1). The goal is to find a good output policy $\hat{\pi}$ with a small sub-optimality gap on the target MDP $\mathcal{M}$, which is measured as:

$$\text{Gap}(\hat{\pi}; \mathcal{M}) := V_1^*(s_1) - V_1^{\hat{\pi}}(s_1).$$

Note that an output policy $\hat{\pi}$ is called $\varepsilon$-optimal if $\text{Gap}(\hat{\pi}; \mathcal{M}) \le \varepsilon$.

**Multiple Data Sources.** The agent has access to datasets from $L$ sources, i.e., $\mathcal{D} := \{\mathcal{D}_l, \forall l \in [L]\}$. Each dataset $\mathcal{D}_l := \{(s_{h,l}^k, a_{h,l}^k, r_{h,l}^k, s_{h+1,l}^k) : k \in [K], h \in [H]\}$ consists of $K$ tuples for each step $h \in [H]$ sampled by a (possibly different) unknown behavior policy $\rho_l$ on a unknown data source MDP denoted as $\mathcal{M}_l = (H, \mathcal{S}, \mathcal{A}, \mathbb{P}_l, r_l)$. In particular, for each step $h \in [H]$, the behavior policy $\rho_l$ performs $K$ times sampling over the state-action space $\mathcal{S} \times \mathcal{A}$ following a distribution denoted as $d_{h,l}^{\rho_l}(\cdot)$, i.e., for each $k \in [K], (s_{h,l}^k, a_{h,l}^k) \sim d_{h,l}^{\rho_l}(\cdot)$. Then, for a sampled pair $(s_{h,l}^k, a_{h,l}^k)$, the reward $r_{h,l}^k = r_{h,l}(s_{h,l}^k, a_{h,l}^k)$ and $s_{h+1,l}^k \sim \mathbb{P}_{h,l}(\cdot|s_{h,l}^k, a_{h,l}^k)$ realized from the date source MDP $\mathcal{M}_l$ is collected and aggregated as a tuple $(s_{h,l}^k, a_{h,l}^k, r_{h,l}^k, s_{h+1,l}^k)$ for the dataset $\mathcal{D}_l$. To ease the presentation, all the sampled tuples are assumed to be independent of each other. Such independence can be ensured by applying the sub-sampling technique in Li et al. (2022) to trajectories induced by behavior policies while maintaining the order of the dataset size.

## 2.2 THE TASK–SOURCE RELATIONSHIP

On one hand, motivated by Sec. 1, each data source MDP $\mathcal{M}_l$ may not exactly match the target task $\mathcal{M}$. Concretely, while having the same episodic length, state space, and action space, their transition and reward dynamics are not necessarily aligned, i.e., $\mathbb{P}_{h,l}(\cdot|s, a) \ne \mathbb{P}_h(\cdot|s, a), r_{h,l}(s, a) \ne r_h(s, a)$.

On the other hand, in practical applications, while being heterogeneous, the data source MDPs are often still related to the target task (e.g., dialogue datasets in Sec. 1). In particular, data sources in offline meta-RL are often assumed to be sampled from a certain distribution (Mitchell et al., 2021). Thus, the following relationship between the target MDP and the data source MDPs is assumed.

**Assumption 1** (Task–source Relationship). *Data source MDPs $\{\mathcal{M}_l : l \in [L]\}$ are independently sampled from an unknown distribution $g$ such that for each $(l, s, a, h) \in [L] \times \mathcal{S} \times \mathcal{A} \times [H]$, the reward $r_{h,l}(s, a)$ is a random variable with mean $r_h(s, a)$, and the transition vector $\mathbb{P}_{h,l}(\cdot|s, a)$ is a random vector with mean $\mathbb{P}_h(\cdot|s, a)$. In particular, random variables and random vectors $\{r_{h,l}(s, a), \mathbb{P}_{h,l}(\cdot|s, a) : (s, a, h, l) \in \mathcal{S} \times \mathcal{A} \times [H] \times [L]\}$ are independent with each other.*

The requirement that rewards are random samples with the expectation as the target model is commonly adopted in bandits literature (Shi & Shen, 2021; Zhu & Kveton, 2022), and the same requirement on the transition vectors is a natural extension, where one representative example is to have them follow a Dirichlet distribution (Marchal & Arbel, 2017).

**Miscellaneous.** Notations without subscripts $l$ generally refer to the target MDP $\mathcal{M}$, while subscripts $l$ are always added when discussing each individual source $\mathcal{M}_l$. For a clear presentation, the notation $c$ is used throughout the paper with varing values to represent a poly-logarithmic term of order $O(\text{poly}(\log(LHSA/\delta)))$ for the tabular setting and $O(\text{poly}(\log(LH/\delta)))$ for the linear setting, where $\delta \in (0, 1)$ is a confidence parameter. Additionally, for $x \in \mathbb{R}$, $x^+$ denotes $\max\{x, 0\}$.

For any function $f : \mathcal{S} \to \mathbb{R}$, the transition operator and Bellman operator of the target MDP $\mathcal{M}$ at each step $h \in [H]$ are defined, respectively, as $(\mathbb{P}_h f)(s, a) := \mathbb{E}[f(s')|s, a]$ and $(\mathbb{B}_h f)(s, a) := r_h(s, a) + (\mathbb{P}_h f)(s, a)$, where the expectation is w.r.t. the transition $s' \sim \mathbb{P}_h(\cdot|s, a)$. Expectation $\mathbb{E}_{\pi, \mathcal{M}}[\cdot]$ is over the trajectory induced by $\pi$ on the MDP $\mathcal{M}$ starting from $s_1$, and $d_h^\pi(s, a)$ denotes the probability of visiting state-action pair $(s, a)$ at step $h$ with policy $\pi$ on $\mathcal{M}$ starting from $s_1$.

---

**Algorithm 1** HetPEVI

---

**Input:** Dataset $\mathcal{D} = \{D_l : l \in [L]\}$; $\hat{V}_{H+1}(s) = 0, \forall s \in \mathcal{S}$
1: For each $(l, s, a, h) \in [L] \times \mathcal{S} \times \mathcal{A} \times [H]$, estimate $\hat{r}_{h,l}(s, a) = r_{h,l}(s, a)\mathbb{1}\{N_{h,l}(s, a) \neq 0\}$
   and $\hat{\mathbb{P}}_{h,l}(s'|s, a) = \frac{N_{h,l}(s, a, s')}{N_{h,l}(s, a) \vee 1}$
2: For each $(s, a, h) \in \mathcal{S} \times \mathcal{A} \times [H]$, aggregate $\hat{r}_h(s, a) = \frac{1}{L} \sum_{l \in [L]} \hat{r}_{h,l}(s, a)$ and $\hat{\mathbb{P}}_h(s'|s, a) = \frac{1}{L} \sum_{l \in [L]} \hat{\mathbb{P}}_{h,l}(s'|s, a)$
3: **for** $h = H, H - 1, \cdots, 1$ **do**
4:     Perform updates for all $(s, a) \in \mathcal{S} \times \mathcal{A}$ as Eqn. (1) with $(\hat{\mathbb{B}}_h \hat{V}_{h+1})(s, a) := \hat{r}_h(s, a) + (\hat{\mathbb{P}}_h \hat{V}_{h+1})(s, a)$ and $\Gamma_h(s, a) = c\sqrt{\sum_{l \in [L]} \frac{H^2}{(L^2 N_{h,l}(s, a)) \vee L}} + c\sqrt{\frac{H^2}{L}}$
5: **end for**
**Output:** policy $\hat{\pi} = \{\hat{\pi}_h(s) : (s, h) \in \mathcal{S} \times [H]\}$

---

## 3  THE HETPEVI ALGORITHM

In this section, the HetPEVI algorithm is introduced for the tabular MDP setting (presented in Algorithm 1). HetPEVI follows the principle of pessimistic value iterations (PEVI) (Rashidinejad et al., 2021; Jin et al., 2021b), which is first briefly introduced, while the key challenges and our novel design in learning the target MDP from randomly perturbed data sources are then illustrated.

The HetPEVI algorithm begins by counting the number of visitations for each tuple $(s, a, h, s') \in \mathcal{S} \times \mathcal{A} \times [H] \times \mathcal{S}$ in each dataset $l \in [L]$. Especially, we denote $N_{h,l}(s, a)$ and $N_{h,l}(s, a, s')$ as the amount of visitations on $(s, a, h)$ and $(s, a, h, s')$ in dataset $\mathcal{D}_l$, respectively. Empirical estimations of rewards and transitions are then given for all $(l, s, a, h) \in [L] \times \mathcal{S} \times \mathcal{A} \times [H]$ as

$$\hat{r}_{h,l}(s, a) = r_{h,l}(s, a)\mathbb{1}\{N_{h,l}(s, a) \neq 0\}; \qquad \hat{\mathbb{P}}_{h,l}(s'|s, a) = \frac{N_{h,l}(s, a, s')}{N_{h,l}(s, a) \vee 1}.$$

These individual estimates are then aggregated into overall estimates as:

$$\hat{r}_h(s, a) = \frac{1}{L} \sum_{l \in [L]} \hat{r}_{h,l}(s, a); \qquad \hat{\mathbb{P}}_h(s'|s, a) = \frac{1}{L} \sum_{l \in [L]} \hat{\mathbb{P}}_{h,l}(s'|s, a).$$

With these estimations, HetPEVI iterates backward from the last step to the first step as

$$\hat{Q}_h(s, a) = \min\left\{ (\hat{\mathbb{B}}_h \hat{V}_{h+1})(s, a) - \Gamma_h(s, a), H - h + 1 \right\}^+,$$
$$\hat{V}_h(s) = \max_{a \in \mathcal{A}} \hat{Q}_h(s, a), \qquad \hat{\pi}_h(s) = \arg\max_{a \in \mathcal{A}} \hat{Q}_h(s, a), \tag{1}$$

with $\hat{V}_{H+1}(s) = 0, \forall s \in \mathcal{S}$, and the empirical Bellman operator $\hat{\mathbb{B}}_h$ is defined as

$$(\hat{\mathbb{B}}_h \hat{V}_{h+1})(s, a) := \hat{r}_h(s, a) + (\hat{\mathbb{P}}_h \hat{V}_{h+1})(s, a),$$

where $(\hat{\mathbb{P}}_h \hat{V}_{h+1})(s, a)$ is the empirical version of $(\mathbb{P}_h \hat{V}_{h+1})(s, a)$ using the estimated $\hat{\mathbb{P}}_h(\cdot|s, a)$.

The essence of the above procedure is that instead of directly setting $\hat{Q}_h(s, a)$ as $(\hat{\mathbb{B}}_h \hat{V}_{h+1})(s, a)$ (as in the standard value iteration), a penalty term $\Gamma_h(s, a)$ is subtracted, which serves the important role of keeping the estimations $\hat{V}_h(s)$ and $\hat{Q}_h(s, a)$ pessimistic. Especially, the penalty term $\Gamma_h(s, a)$ should be carefully designed such that with a high probability, it holds that

$$\left|(\hat{\mathbb{B}}_h \hat{V}_{h+1})(s, a) - (\mathbb{B}_h \hat{V}_{h+1})(s, a)\right| \leq \Gamma_h(s, a), \qquad \forall(s, a, h) \in \mathcal{S} \times \mathcal{A} \times [H]. \tag{2}$$

**Technical Challenges.** Previous offline RL studies (Jin et al., 2021b; Rashidinejad et al., 2021; Yin et al., 2022; Xie et al., 2021b) only deal with one single target data source and only one type

of uncertainties (finite data samples) to ensure Eqn. (2). In this work, the agent needs to process multiple heterogeneous datasets, while none of them individually characterizes the learning target. As a result, the agent faces two *coupled* challenges. First, the uncertainties due to the finite sample sizes still needs to be considered. This is referred to as the **sample uncertainties**, but the key difference is that now the agent needs to jointly consider and aggregate uncertainties associated with all data sources. Second, even with perfect knowledge of each data source, the target MDP may not be fully revealed. Thus, the agent also needs to consider the uncertainties from the limited number of data sources, which are referred to as the **source uncertainties**.

To address the two uncertainties, the penalty term is designed to have two parts as follows:

$$\Gamma_h(s,a) = \Gamma_h^\alpha(s,a) + \Gamma_h^\beta(s,a),$$

where $\Gamma_h^\alpha(s,a)$ aggregates the sample uncertainties from each data source while $\Gamma_h^\beta(s,a)$ accounts for the source uncertainties. Both of them are further elaborated on in the following.

**Penalties to Aggregate Sample Uncertainties.** For the first part of the penalty, i.e., $\Gamma_h^\alpha(s,a)$, since the agent faces data from multiple heterogeneous sources, the penalty term needs to jointly consider the individual uncertainties from all sources. In HetPEVI, the following penalty term is proposed, which originates from the Hoeffding inequality:

$$\Gamma_h^\alpha(s,a) = c\sqrt{\sum_{l\in[L]} \frac{H^2}{(L^2 N_{h,l}(s,a)) \vee L}} + c\sqrt{\frac{H^2}{L}}.$$

Note that instead of treating each source individually and directly summing up their sample uncertainties (as $c\sum_{l\in[L]} \frac{1}{L}\sqrt{\frac{H^2}{N_{h,l}(s,a)\vee 1}}$), the adopted penalty term is a joint measure of sample uncertainties from all sources. This would lead to a $O(\sqrt{L})$ speed-up in the performance guarantees.

**Penalties to Account for Source Uncertainties.** The second part of the penalty $\Gamma_h^\beta(s,a)$ serves the important role of measuring the uncertainties due to the limited amount of data sources. With the observation that $(\mathbb{B}_{h,l}\hat{V}_{h+1})(s,a)$ is a bounded random variable with mean of $(\mathbb{B}_h\hat{V}_{h+1})(s,a)$, the following penalty term is proposed:

$$\Gamma_h^\beta(s,a) = c\sqrt{\frac{H^2}{L}}.$$

Intuitively, it shrinks with the number of datasets, i.e., $L$, as more sources provide more information about the target task.

*Remark* 1. With the two-part penalty term, with high probability, for all $(s,a,h) \in \mathcal{S} \times \mathcal{A} \times [H]$, it simultaneously holds that $|(\hat{\bar{\mathbb{B}}}_h\hat{V}_{h+1})(s,a) - (\bar{\mathbb{B}}_h\hat{V}_{h+1})(s,a)| \leq \Gamma_h^\alpha(s,a)$ and $|(\bar{\mathbb{B}}_h\hat{V}_{h+1})(s,a) - (\mathbb{B}_h\hat{V}_{h+1})(s,a)| \leq \Gamma_h^\beta(s,a)$, which jointly ensure Eqn. (2).

*Remark* 2. The adopted source uncertainty penalty $\Gamma_h^\beta(s,a)$ is intended to accommodate any unknown variance between sources. However, if there is prior knowledge of the variance, it is feasible to incorporate such information. In particular, if the rewards and transition vectors are generated via $\sigma$-sub-Gaussian distributions, the penalty can be designed as $\Gamma_h^\beta(s,a) = c\sqrt{\sigma^2 H^2/L}$.

## 4 THEORETICAL ANALYSIS

### 4.1 INDIVIDUALLY GOOD COVERAGE

Previous offline RL studies typically require that the behavior policy should provide information for all the state-action pairs that may be visited by the optimal policy on the target MDP, e.g., Rashidinejad et al. (2021); Xie et al. (2021b). Particularizing this requirement to each individual data source considered in this work leads to the following assumption, which requires each individual dataset covers the optimal policy on the target MDP.

**Assumption 2** (Individual Coverage). *There exists a constant $C^* < \infty$ such that for all $(s,a,h,l) \in \mathcal{S} \times \mathcal{A} \times [H] \times [L]$, it holds that $d_h^{\pi^*}(s,a) \leq C^* d_{h,l}^{\rho_l}(s,a)$.*

Under this assumption, a minimax lower bound is first established as follows.

**Theorem 1** (Lower Bound; Individual Coverage). *For any $C^* \geq 2, S \geq 2, A \geq 2$, it holds that*

$$\inf_{\hat{\pi}} \sup_{\mathcal{M} \in \mathfrak{M}, g \in \mathfrak{G}, \{\rho_l : l \in [L]\} \in \mathfrak{B}(C^*)} \mathbb{E}_{\{\mathcal{M}_l, \mathcal{D}_l : l \in [L]\}} \left[ Gap(\hat{\pi}; \mathcal{M}) \right] = \Omega \left( \sqrt{\frac{C^* H^3 S}{LK}} \vee \sqrt{\frac{H^2}{L}} \right),$$

*where $\mathfrak{M} := \{\mathcal{M}(H, \mathcal{S}, \mathcal{A}, \mathbb{P}, r)\}$ is the family of all possible target MDPs, $\mathfrak{G} := \{g: \{\mathcal{M}_l \sim g : l \in [L]\}$ satisfies Assumption 1\} is a family of data source generation distributions, and $\mathfrak{B}(C) := \{\{\rho_l : l \in [L]\} : Assumption 2 holds with $C^* = C\}$ is a family of behavior policies.*

The first term in this lower bound is the same as the lower bound of standard offline RL with $LK$ data samples directly from the target MDP (Rashidinejad et al., 2021), which represents the performance loss originating from finite data samples. The second term is unique for the setting studied in this work, as it represents the loss caused by learning with finite randomly perturbed data sources. Especially, it can be observed that the second term only scales with the number of data sources, i.e., $L$, and cannot be mitigated via sampling more data from each data source, i.e., increasing $K$. On one hand, this observation verifies the intuition that using data sampled from multiple randomly perturbed data sources poses additional learning difficulties. On the other hand, it also highlights the importance of the diversity of data sources, i.e., it is more important to involve more sources instead of more data from each source. This is reasonable as involving more data sources provides additional population coverage while also adding more data.

With the information-theoretic lower bound established, the performance guarantee of HetPEVI is also provided in the following theorem, which highlights its effectiveness and efficiency.

**Theorem 2** (HetPEVI; Individual Coverage). *Under Assumptions 1, 2, w.p. at least $1 - \delta$, the output policy $\hat{\pi}$ of HetPEVI satisfies*

$$Gap(\hat{\pi}; \mathcal{M}) = \tilde{O} \left( \sqrt{\frac{C^* H^4 S}{LK}} + \sqrt{\frac{H^4}{L}} \right).$$

This result first illustrates that even with finite randomly perturbed MDPs and finite data samples from each of them, it is still feasible to efficiently learn the target. Compared with the lower bound in Theorem 1, it can be observed that HetPEVI is optimal (up to logarithmic factors) on the dependency of $L, K, C^*$ and $S$. The additional $\sqrt{H}$ factor in the first term (from the sample uncertainties) can be removed by invoking a carefully designed Bernstein-type penalty term to incorporate the variance information, which is deferred to Appendix E due to the space limitation. However, it is currently unclear how to alleviate the additional $H$ factor in the second term (from the source uncertainties), which is left open to be further investigated.

Additionally, Thm. 2 indicates that to obtain an $\varepsilon$-optimal policy, HetPEVI requires the overall number of samples $T = LK$ to be of order $\tilde{O}(C^* H^4 S / \varepsilon^2)$ and the available number of data sources $L$ to be of order $\tilde{O}(H^4 / \varepsilon^2)$. Note that the first requirement can be viewed as on the *sample complexity* while the second one is on the *source diversity*.

## 4.2 COLLECTIVELY GOOD COVERAGE

It can be noticed that Assumption 2 requires *each* data source to provide good coverage. With access to only one data source in previous offline RL studies, this requirement is intuitive as information about the optimal policy needs to be obtained. However, the multiple available datasets in this work provide opportunities to avoid this strong assumption via their aggregated information. In this section, the scenario where the datasets collectively provide a good coverage is discussed. In particular, to characterize the collective coverage, the following assumption is proposed.

**Assumption 3** (Collective Coverage). *There exist constants $L^\dagger > 0$ and $C^\dagger < \infty$ such that for all $(s, a, h) \in \mathcal{S} \times \mathcal{A} \times [H]$, it holds that $|\{l \in [L] : d_h^{\pi^*}(s, a) \leq C^\dagger d_{h,l}^{\rho_l}(s, a)\}| \geq L^\dagger$.*

It can be observed that Assumption 3 is weaker than Assumption 2 as the latter implies the former with $L^\dagger = L$ and $C^\dagger = C^*$. Moreover, instead of requiring each data source individually covers the optimal policy, Assumption 3 leverages their collective coverage. In particular, for each different

state-action pair possibly visited by the optimal policy (i.e., $d_h^{\pi^*}(s,a) > 0$), it is sufficient to have potentially varying datasets cover it. In other words, different parts of the optimal policy can be covered by different datasets, which is a highly practical consideration. Under Assumption 3, the following performance guarantee of HetPEVI can be further obtained.

**Theorem 3** (HetPEVI; Collective Coverage). *Under Assumptions 1, 3, w.p. at least $1-\delta$, the output policy $\hat{\pi}$ of HetPEVI satisfies*

$$Gap(\hat{\pi}; \mathcal{M}) = \tilde{O}\left(\sqrt{\frac{C^\dagger H^4 S}{LK}} + \sqrt{\frac{(L+1-L^\dagger)H^4}{L}}\right).$$

Compared with Theorem 2, Theorem 3 is more general and also more practically useful as it implies the former and indicates that good collective coverage is sufficient for efficient learning. Moreover, from another perspective, the performance dependence on $L^\dagger$ also highlights the importance of collecting high-quality datasets.

As a summary of the obtained observations, learning via randomly perturbed datasets brings both costs and benefits. Especially, an unavoidable performance loss occurs due to the indirect access to the target MDP. This loss can only be reduced by increasing the diversity of data sources, i.e., a larger $L$, but cannot be mitigated by collecting a larger dataset from each data source, i.e., a larger $K$. Despite bringing the additional loss, the access to multiple heterogeneous datasets provides an opportunity to leverage their aggregated information, which is concretely reflected that efficient learning only requires a good collective (instead of individual) coverage.

## 5 EXTENSION TO OFFLINE LINEAR MDP

Lastly, we extend the study from tabular RL to incorporating function approximations. Especially, the following linear MDP model is considered.

**Definition 1** (Linear MDP (Jin et al., 2020; 2021b)). *An MDP $\mathcal{M} = (H, \mathcal{S}, \mathcal{A}, \mathbb{P}, r)$ is a linear MDP with a feature map $\phi : \mathcal{S} \times \mathcal{A} \to \mathbb{R}^d$ if there exist $d$ unknown measures $\mu_h = (\mu_h^{(1)}, \cdots, \mu_h^{(d)})$ over $\mathcal{S}$ and an unknown vector $\theta_h \in \mathbb{R}^d$ s.t. $\mathbb{P}_h(s'|s,a) = \langle \phi(s,a), \mu_h(s') \rangle$ and $r_h(s,a) = \langle \phi(s,a), \theta_h \rangle$ for all $(s,a,s') \in \mathcal{S} \times \mathcal{A} \times \mathcal{S}$ at each step $h \in [H]$. Without loss of generality, it is assumed that $\|\phi(s,a)\|_2 \leq 1$ for all $(s,a) \in \mathcal{S} \times \mathcal{A}$ and $\max\{\|\mu_h(\mathcal{S})\|_2, \|\theta_h\|_2\} \leq \sqrt{d}$ for all $h \in [H]$, where $\|\mu_h(\mathcal{S})\|_2 := \int_{\mathcal{S}} \|\mu_h(s)\|_2 \,\mathrm{d}s$. For simplicity, we denote $\mathcal{M} = (H, \mathcal{S}, \mathcal{A}, \mu, \phi, \theta)$.*

With this definition, we consider the problem that the target MDP $\mathcal{M}$ is a linear MDP, i.e., $\mathcal{M} = (H, \mathcal{S}, \mathcal{A}, \phi, \mu, \theta)$. Correspondingly, the data source MDPs $\{\mathcal{M}_l : l \in [L]\}$ are also assumed to be linear MDPs, which are denoted as $\{\mathcal{M}_l = (H, \mathcal{S}, \mathcal{A}, \phi, \mu_l, \theta_l) : l \in [L]\}$. In particular, the data source MDPs are assumed to share the same feature dimension $d$ and feature mapping $\phi$ as the target MDP; however, their system dynamics may be different. We note that this shared feature mapping is commonly adopted in federated linear bandits (Huang et al., 2021; Li & Wang, 2022), which naturally extends to linear MDP. Then, similarly as Assumption 1 in the tabular MDP, the following task-source relationship is assumed for the linear MDP to model that the data sources are randomly perturbed versions of the target MDP.

**Assumption 4** (Task–source Relationship; Linear MDP). *Data source MDPs $\{\mathcal{M}_l : l \in [L]\}$ are independently sampled from unknown distributions $g$ such that for each $(l, i, h) \in [L] \times [d] \times [H]$, the vector $\theta_{h,l}$ is a random vector with mean $\theta_h$, and the measure $\mu_{h,l}^{(i)}$ is a random measure with mean $\mu_h^{(i)}$. In particular, random vectors and measures $\{\theta_{h,l}, \mu_{h,l}^{(i)} : (h, i, l) \in [H] \times [d] \times [L]\}$ are independent with each other.*

Note that when treating the tabular setting as $d = SA$ and $\phi(s,a) = \mathbf{e}_{(s,a)}$, where $\mathbf{e}_{(s,a)}$ denotes a canonical basis in $\mathbb{R}^d$, Assumption 4 returns to Assumption 1.

### 5.1 HETPEVI-LIN FOR LINEAR MDPS

From the linear MDP definition, it can be shown that the Bellman operators of each data source MDP and the target MDP are all linear (but with different weights).

---

**Algorithm 2** HetPEVI-Lin

---

**Input:** Dataset $\mathcal{D} = \{D_l : l \in [L]\}$; $\hat{V}_{H+1}(s) = 0, \forall s \in \mathcal{S}$

1: Estimate $\hat{w}_{h,l} \leftarrow \Lambda_{h,l}^{-1} \left[ \sum_{k \in [K]} \phi(s_{h,l}^k, a_{h,l}^k) \left( r_{h,l}^k + \hat{V}_{H+1}(s_{h+1,l}^k) \right) \right], \forall (h,l) \in [H] \times [L]$

2: Set $\hat{w}_h \leftarrow \sum_{l \in [L]} \hat{w}_{h,l}/L, \forall h \in [H]$

3: **for** $h = H, H-1, \cdots, 1$ **do**

4:     Perform updates for all $(s,a) \in \mathcal{S} \times \mathcal{A}$ as Eqn. (1) with $(\hat{\mathbb{B}}_h \hat{V}_{H+1})(s,a) = \hat{w}_h^\top \phi(s,a)$ and
$\Gamma_h(s,a) = c\sqrt{\frac{dH^2}{L^2}} \sqrt{\sum_{l \in [L]} \|\phi(s,a)\|_{\Lambda_{h,l}^{-1}}^2} + c\sqrt{\frac{dH^2}{L}}$

5: **end for**

**Output:** policy $\hat{\pi} = \{\hat{\pi}_h(s) : (s,h) \in \mathcal{S} \times [H]\}$

---

**Lemma 1.** *With any function $f : \mathcal{S} \to \mathbb{R}$, there exists $\{w_{h,l}^f \in \mathbb{R}^d : h \in [H], l \in [L]\}$ and $\{w_h^f \in \mathbb{R}^d : h \in [H]\}$ such that for any $(s,a,h) \in \mathcal{S} \times \mathcal{A} \times [H]$, it holds that $(\mathbb{B}_{h,l}f)(s,a) = \langle \phi(s,a), w_{h,l}^f \rangle$ for each $l \in [L]$ and $(\mathbb{B}_h f)(s,a) = \langle \phi(s,a), w_h^f \rangle$.*

Furthermore, the following lemma can be established for the sample average MDP $\bar{\mathcal{M}} = (H, \mathcal{S}, \mathcal{A}, \bar{\mathbb{P}}, r)$, where $\bar{\mathbb{P}}_h(\cdot|s,a) = \frac{1}{L}\sum_{l \in [L]} \mathbb{P}_{h,l}(\cdot|s,a)$ and $\bar{r}_h(s,a) = \frac{1}{L}\sum_{l \in [L]} r_{h,l}(s,a)$. It states that $\bar{\mathcal{M}}$ is also a linear MDP and the weights associated with its Bellman operator are the average of weights from each individual data source MDP.

**Lemma 2.** *The average MDP $\bar{\mathcal{M}}$ is a linear MDP of dimension $d$ with feature mapping $\phi$. Also, for any function $f : \mathcal{S} \to \mathbb{R}$ and any $(s,a,h) \in \mathcal{S} \times \mathcal{A} \times [H]$, it holds that $(\bar{\mathbb{B}}_h f)(s,a) = \langle \phi(s,a), \bar{w}_h^f \rangle$ with weights $\{\bar{w}_h^f = \sum_{l \in [L]} w_{h,l}^f/L : h \in [H]\}$.*

With the above observations, the HetPEVI-Lin algorithm is proposed (presented in Algorithm 2). First, for each step $h \in [H]$ and each data source $l \in [L]$, the weight $\hat{w}_{h,l}$ is estimated via a ridge regression:

$$\hat{w}_{h,l} = \arg\min_{w \in \mathbb{R}^d} \sum_{k \in [K]} \left[ r_{h,l}^k + \hat{V}_{h+1}(s_{h+1,l}^k) - \phi(s_{h,l}^k, a_{h,l}^k)^\top w \right]^2 + \frac{1}{L}\|w\|_2^2$$

$$= \Lambda_{h,l}^{-1}\left[ \sum_{k \in [K]} \phi(s_{h,l}^k, a_{h,l}^k)\left(r_{h,l}^k + \hat{V}_{h+1}(s_{h+1,l}^k)\right)\right],$$

with $\Lambda_{h,l} := \sum_{k \in [K]} \phi(s_{h,l}^k, a_{h,l}^k)\phi(s_{h,l}^k, a_{h,l}^k)^\top + I/L$. The agent aggregates $\hat{w}_h = \sum_{l \in [L]} \hat{w}_{h,l}/L$, which provides an estimation of the weight $\bar{w}_h$ associated with the average MDP $\bar{\mathcal{M}}$. The PEVI in Eqn. (1) is then performed with the empirical Bellman operator defined as $(\hat{\mathbb{B}}_h \hat{V}_{h+1})(s,a) = \langle \phi(s,a), \hat{w}_h \rangle$ and a two-part penalty term $\Gamma_h(s,a) = \Gamma_h^\alpha(s,a) + \Gamma_h^\beta(s,a)$ illustrated in the following.

**Penalties to Aggregate Sample Uncertainties.** The first part of the penalty term aggregates the sample uncertainties from each data source, which is designed as

$$\Gamma_h^\alpha(s,a) = c\sqrt{\frac{dH^2}{L^2}} \sqrt{\sum_{l \in [L]} \|\phi(s,a)\|_{\Lambda_{h,l}^{-1}}^2}.$$

While this penalty term shares a similar format as those in linear bandits (Li et al., 2010; Abbasi-Yadkori et al., 2011) and linear MDPs (Jin et al., 2020; 2021b), its novelty comes from jointly considering the uncertainties of different sources. Especially, this design avoids applying the self-normalized concentration (Abbasi-Yadkori et al., 2011) to each data source (which leads to an inferior dependency on $L$) by leveraging the statistical independence among the split datasets.

**Penalties to Account for Source Uncertainties.** Besides the sample uncertainties, the following penalty is designed to measure the source uncertainties:

$$\Gamma_h^\beta(s,a) = c\sqrt{\frac{dH^2}{L}}.$$

Note that compared with the tabular design, an additional $d$ factor appears, which is from the covering argument over the $d$-dimensional feature space.

## 5.2 THEORETICAL ANALYSIS

Similar to the tabular setting, previous offline RL studies with linear MDPs often requires the behavior policy to cover the optimal policy (Jin et al., 2021b; Xiong et al., 2022) (or even stronger, to cover the entire feature space (Yin et al., 2022; Min et al., 2021)). Hence, following Sec. 4, we first discuss the scenario with good individual coverage characterized by the following assumption.

**Assumption 5** (Individual Coverage; Linear MDP). *There exists a constant $D^* < \infty$ such that for all $(h, l) \in [H] \times [L]$, it holds that $\mathbb{E}_{\pi^*, \mathcal{M}}[\phi(s_h, a_h)\phi(s_h, a_h)^\top] \preceq D^* \mathbb{E}_{\rho_l, \mathcal{M}_l}[\phi(s_h, a_h)\phi(s_h, a_h)^\top].$*

**Theorem 4** (HetPEVI-Lin; Individual Coverage). *Under Assumptions 4, 5 and assuming the matrices $\{\sum_{l \in [L]} \Lambda_{h,l}^{-1} : h \in [H]\}$ are invertible, w.p. at least $1 - \delta$, the output policy $\hat{\pi}$ of HetPEVI-Lin satisfies*

$$Gap(\hat{\pi}; \mathcal{M}) = \tilde{O}\left(\sqrt{\frac{D^* d^2 H^4}{KL}} + \sqrt{\frac{dH^4}{L}}\right).$$

Similar to Thm. 2, the two terms in Thm. 4 originate from the finite data samples and the limited data sources, respectively. However, note that the gap guarantee in Thm. 4 does not have a dependency on the number of states $S$ (which appears in the tabular analysis), instead it depends on the feature dimension $d$, thanks to the careful design that incorporates linear function approximation. As a result, to output an $\varepsilon$-optimal policy, the sample complexity requirement is $T = KL = \tilde{O}(D^* d^2 H^4 / \varepsilon^2)$ while the source diversity requirement is $L = \tilde{O}(dH^4 / \varepsilon^2)$.

One important observation from the tabular setting is that efficient learning only requires a good collective (instead of individual) coverage. To further verify this claim, the following collective coverage assumption is considered for linear MDPs, which shares a similar format as Assumption 3.

**Assumption 6** (Collective Coverage; Linear MDP). *There exist constants $L^\dagger > 0$ and $D^\dagger < \infty$ such that for all $h \in [H]$, it holds that $|l \in [L] : \mathbb{E}_{\pi^*, \mathcal{M}}[\phi(s_h, a_h)\phi(s_h, a_h)^\top] \preceq D^* \mathbb{E}_{\rho_l, \mathcal{M}_l}[\phi(s_h, a_h)\phi(s_h, a_h)^\top]| \geq L^\dagger.$*

It can be observed that Assumption 5 implies Assumption 6 with $L^\dagger = L$ and $D^\dagger = D^*$. With this relatively weaker collective coverage assumption, the following theorem can be established.

**Theorem 5** (HetPEVI-Lin; Collective Coverage). *Under Assumptions 4, 6 and assuming the matrices $\{\sum_{l \in [L]} \Lambda_{h,l}^{-1} : h \in [H]\}$ are invertible, w.p. at least $1 - \delta$, the output policy $\hat{\pi}$ of HetPEVI-Lin satisfies*

$$Gap(\hat{\pi}; \mathcal{M}) = \tilde{O}\left(\sqrt{\frac{D^\dagger d^2 H^4}{KL}} + \sqrt{\frac{d(L + 1 - L^\dagger)H^4}{L}}\right).$$

It can be observed that Thm. 3 shares a similar form as Thm. 3, both of which illustrate that as long as the datasets collectively cover the optimal policy, efficient learning is achievable.

## 6 CONCLUSIONS

This work studied a novel problem of efficient offline RL with randomly perturbed data sources. In particular, motivated by practical applications, the available offline datasets are assumed to be collected from multiple randomly perturbed versions of the target MDP. The HetPEVI algorithm is proposed, where novel penalty terms were designed to jointly consider the uncertainties from the finite data samples and the limited amount of data sources. Theoretical analyses proved its near-optimal performance. More importantly, the costs and benefits of offline RL via randomly perturbed data sources are explicitly characterized. On one hand, an additional unavoidable performance loss occurs due to the finite data sources, which cannot be reduced by involving more data samples from each source. On the other hand, as long as the datasets collectively (instead of individually) provide a good data coverage, efficient learning is achievable. Lastly, linear function approximation was considered, and the HetPEVI-Lin algorithm was developed with penalties carefully designed to consider both uncertainties and incorporate the linear structure. Its analysis again verifies the importance of source diversity and the sufficiency of collective coverage.

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

# A    ADDITIONAL DISCUSSIONS

## A.1    RELATED WORKS

RL has seen much progress over the past few years, particularly in its theoretical understanding (see the recent monograph (Agarwal et al., 2019) for an overview). We will discuss the most related papers in the following, with a particular focus on the theoretical aspect as well as the offline setting.

**Online RL with One Environment.** The majority of RL studies focus on the online setting (Sutton & Barto, 2018), where the agent consistently improves her algorithm via online interactions with the environment. In recent years, many advances have deepened the theoretical understandings of online RL in interacting with a fixed and stable environment, with provably efficient designs (Azar et al., 2017; Jin et al., 2018; 2020; 2021a; Zhang et al., 2020; Jiang et al., 2017; Dann et al., 2021).

**Online RL with Heterogeneous Environments.** The importance of handling heterogeneous environments is well recognized. Especially, federated RL (Zhuo et al., 2019) and the general multi-agent RL (Zhang et al., 2021) study the setting with multiple agents in the system, while each of them may view an agent-depend environment (Jin et al., 2022). In contrast, meta-RL (Sæmundsson et al., 2018; Gupta et al., 2018; Fallah et al., 2021; Chua et al., 2021) focuses on extracting knowledge from previous tasks and adapting them to new (potentially different) tasks. Also, multi-task RL (Teh et al., 2017) attempts to handle multiple tasks together via information shared among tasks (Brunskill & Li, 2013; Zhang & Wang, 2021; Lu et al., 2021; Yang et al., 2020; 2022) or distinguishing the latent structure associated with each task via past observations (Hallak et al., 2015; Kwon et al., 2021a;b).

**Offline RL with Single-source Datasets.** Offline RL (Levine et al., 2020) attempts to avoid potentially expensive interactions with the environment as in online RL but instead use available offline datasets collected previously. Inspired by empirical advances (Yu et al., 2020; Kumar et al., 2020), the principle of "pessimism" is incorporated and proved efficient for offline RL (Jin et al., 2021b; Rashidinejad et al., 2021). Especially, it is illustrated that a nearly optimal policy can be learned via a dataset collected by a behavior policy that covers the trajectories of the optimal policy.

Following this line, Xie et al. (2021b); Li et al. (2022); Shi et al. (2022) further fine-tune the designs for the tabular setting, and Jin et al. (2021b); Zanette et al. (2021); Min et al. (2021); Yin et al. (2022); Xiong et al. (2022) for the linear MDP. For the general function approximation, additional attempts are reported in Xie et al. (2021a); Uehara & Sun (2021). However, these results are mostly information-theoretical as computational intractable optimizations are required in general.

Besides the above discussions, one particular line of research that relates to this work is under the category of offline robust RL (Zhou et al., 2021; Yang et al., 2021; Panaganti et al., 2022; Panaganti & Kalathil, 2022; Si et al., 2020; Shi & Chi, 2022). Especially, offline robust RL learns via data sampled from *one* MDP and tries to output a policy that is robustly good for a family of MDPs; however, our work attempts at learning the hidden task from data sampled from a family of MDPs. Nevertheless, we note that it would be an interesting question to investigate whether having access to multiple data source MDPs in the target family would make offline robust RL easier.

Another conceptually relative work is Shrestha et al. (2020). In particular, it looks for similar state-action pairs with small distances in the dataset, which can be thought of as available data sources in this work. Then, the Lipschitz continuity assumption is posed, which serves a similar role as Assumption 1 to establish the connection between desired task information with the available datasets. From this perspective, the first term in Theorem 3.1 (Shrestha et al., 2020) can be interpreted as coming from the source uncertainty while the second term is from the sample uncertainty. However, we also note that the Lipschitz continuity assumption is a worst-case consideration that would not characterize the concentration of involving more data sources, which however is the key of this work.

**Offline RL with Multi-source Datasets.** The aforementioned advances on offline RL are mainly focused on learning from a single data source, which limits their applicability. In the offline domain, growing interests have been made to utilize data from heterogeneous sources. The most related literature falls under the framework of "offline meta-RL" or "offline multi-task RL" (Mitchell et al., 2021; Dorfman et al., 2021; Lin et al., 2022; Li et al., 2020b;a; Yu et al., 2021). Especially, the target MDP can be viewed as a learning target for the "meta-training" process of offline meta-

RL (Mitchell et al., 2021), which aims to extract information from the available data (of multiple sources). In addition to "meta-training", the empirically studied offline meta-RL systems often feature another step of "meta-testing", which further utilizes the learned information and applies them to a specific task. Thus, we believe this work may contribute the (currently lacking) theoretical understanding of offline meta-RL systems, especially the meta-training process, which may also serve as the foundation for studies on the meta-testing process.

### A.2 FUTURE WORKS

**Coverage Assumptions.** While the collective coverage assumption of Assumptions 3 and 6 is relatively weak, it is still of major interest to further explore how to perform offline RL (especially with heterogeneous data sources) under weaker assumptions. This direction is particularly interesting with multiple data sources since the heterogeneous sources naturally enrich the data diversity.

**Unknown Source Identities.** This work considers the scenario where each data sample is known to belong to a particular source. One interesting direction is to investigate the scenarios without such information, i.e., unknown source identities. A potential solution is to first cluster the data samples and then adopt the algorithms proposed in this work. However, it is challenging to design provable clustering algorithms. One candidate clustering technique is developed in Kwon et al. (2021b) for the study of latent MDP, which however relies on strong assumptions of prior knowledge about the source MDPs.

**Personalization.** As mentioned in the discussions of related work, this work can be viewed as targeting at the "meta-training" process of offline meta-RL (Mitchell et al., 2021), which extracts common knowledge from available data of multiple sources. While the extracted common knowledge have individual values, in many applications, an additional step of personalization is performed to further use such knowledge to benefit a specific task, which is called the "meta-testing" process of offline meta-RL (Mitchell et al., 2021). Based on this work, it would be valuable to further study how to perform such a personalization step with theoretical guarantees.

## B LOWER BOUND ANALYSIS

**Lemma 3.** *For any $C^* \geq 2$, it holds that*

$$\inf_{\hat{\pi}} \sup_{\mathcal{M} \in \mathfrak{M}, g \in \mathfrak{G}, \{\rho_l : l \in [L]\} \in \mathfrak{B}(C^*)} \mathbb{E}_{\{\mathcal{M}_l, \mathcal{D}_l : l \in [L]\}} \left[ Gap(\hat{\pi}; \mathcal{M}) \right] = \Omega\left( \sqrt{\frac{C^* S H^3}{LK}} \right).$$

*Proof.* We consider the case where $g$ always generates $\mathcal{M}_l = \mathcal{M}$, and $\rho_1 = \cdots = \rho_L = \rho$. Then, the problem degenerates to offline RL with one dataset directly from the target. Thus, results from the case of $C^* \geq 2$ in Theorem 7 of Rashidinejad et al. (2021) can be applied to obtain the final result. □

**Lemma 4.** *For any $C^* \geq 2$, it holds that*

$$\inf_{\hat{\pi}} \sup_{\mathcal{M} \in \mathfrak{M}, g \in \mathfrak{G}, \{\rho_l : l \in [L]\} \in \mathfrak{B}(C^*)} \mathbb{E}_{\{\mathcal{M}_l, \mathcal{D}_l : l \in [L]\}} \left[ Gap(\hat{\pi}; \mathcal{M}) \right] = \Omega\left( \sqrt{\frac{H^2}{L}} \right).$$

*Proof.* This lemma is established with the following construction.

**Target MDP $\mathcal{M}$.** We design the following family of target MDPs with two states denoted as $\mathcal{S} = \{s_g, s_b\}$ and two actions denotes as $\mathcal{A} = \{a_g, a_b\}$ for all steps ($s_1$ can be fixed to be $s_g$), which can be easily generalized to accommodate any number of states and actions:

$$\begin{aligned} \mathfrak{M} = \big\{ &\mathbb{P}_1(s_g|s_g, a_g) = p, \mathbb{P}_h(s_b|s_g, a_g) = 1 - p \\ &\mathbb{P}_1(s_g|s_g, a_b) = \mathbb{P}_1(s_b|s_g, a_b) = 0.5, \forall h \in [H]; \\ &\mathbb{P}_h(s|s, a) = 1, \forall (s, a, h) \in \mathcal{S} \times \mathcal{A} \times [2, H]; \\ &r_h(s_g, a) = 1, r_h(s_b, a) = 0, \forall (a, h) \in \mathcal{A} \times [H] \big\}. \end{aligned}$$

A target MDP $\mathcal{M}$ with parameter $p$ in $\mathfrak{M}$ is referred to as $\mathcal{M}(p)$.

**Data source generation distribution $g$.** For target $\mathcal{M}(p)$, the data source generation, denoted as $g(p)$, is designed as follows: with probability $p$, the generated MDP has $\mathbb{P}_1(s_g|s_g, a_g) = 1, \mathbb{P}_1(s_b|s_g, a_g) = 0$; otherwise it has $\mathbb{P}_1(s_g|s_g, a_g) = 0, \mathbb{P}_1(s_b|s_g, a_g) = 1$. The other parameters of the generated MDP are the same as $\mathcal{M}$.

**Behavior policy $\rho_l$.** For all $l \in [L]$, the behavior policy $\rho_l$ is specified as follows: $\rho_{h,l}(a|s) = d_h^{\pi^*}(a|s), \forall (s, a, h) \in \mathcal{S} \times \mathcal{A} \times [H]$, which ensures Assumption 2 with any $C^* \geq 1$.

Then, with $p_1 := \frac{1}{2} + \delta$ and $p_2 = \frac{1}{2} - \delta$, it can be obtained that

$$\mathbb{E}_{\{\mathcal{M}_l \sim g(p_1), \mathcal{D}_l \sim \mathcal{M}_l : l \in [L]\}} \left[ \text{Gap}(\hat{\pi}; \mathcal{M}(p_1)) \right] = (H-1)\delta \mathbb{E}_{\{\mathcal{M}_l \sim g(p_1), \mathcal{D}_l \sim \mathcal{M}_l : l \in [L]\}} \left[ \pi_1(a_b) \right];$$

$$\mathbb{E}_{\{\mathcal{M}_l \sim g(p_2), \mathcal{D}_l \sim \mathcal{M}_l : l \in [L]\}} \left[ \text{Gap}(\hat{\pi}; \mathcal{M}(p_2)) \right] = (H-1)\delta \mathbb{E}_{\{\mathcal{M}_l \sim g(p_2), \mathcal{D}_l \sim \mathcal{M}_l : l \in [L]\}} \left[ \pi_h(a_g) \right],$$

which leads to

$$\mathbb{E}_{\{\mathcal{M}_l \sim g(p_1), \mathcal{D}_l \sim \mathcal{M}_l : l \in [L]\}} \left[ \text{Gap}(\hat{\pi}; \mathcal{M}(p_1)) \right] + \mathbb{E}_{\{\mathcal{M}_l \sim g(p_2), \mathcal{D}_l \sim \mathcal{M}_l : l \in [L]\}} \left[ \text{Gap}(\hat{\pi}; \mathcal{M}(p_2)) \right]$$

$$= (H-1)\delta \left( \mathbb{E}_{\{\mathcal{M}_l \sim g(p_1), \mathcal{D}_l \sim \mathcal{M}_l : l \in [L]\}} \left[ \pi_1(a_b) \right] + \mathbb{E}_{\{\mathcal{M}_l \sim g(p_2), \mathcal{D}_l \sim \mathcal{M}_l : l \in [L]\}} \left[ 1 - \pi_1(a_g) \right] \right)$$

Furthermore, it holds that

$$\mathbb{E}_{\{\mathcal{M}_l \sim g(p_1), \mathcal{D}_l \sim \mathcal{M}_l : l \in [L]\}} \left[ \pi_1(a_b) \right] + \mathbb{E}_{\{\mathcal{M}_l \sim g(p_2), \mathcal{D}_l \sim \mathcal{M}_l : l \in [L]\}} \left[ 1 - \pi_1(a_g) \right]$$

$$\geq 1 - \text{TV}(\mathbb{P}_{\{\mathcal{M}_l \sim g(p_1), \mathcal{D}_l \sim \mathcal{M}_l : l \in [L]\}}, \mathbb{P}_{\{\mathcal{M}_l \sim g(p_2); \mathcal{D}_l \sim \mathcal{M}_l : l \in [L]\}})$$

$$\geq 1 - \sqrt{\text{KL} \left( \mathbb{P}_{\{\mathcal{M}_l \sim g(p_1), \mathcal{D}_l \sim \mathcal{M}_l : l \in [L]\}} || \mathbb{P}_{\{\mathcal{M}_l \sim g(p_2), \mathcal{D}_l \sim \mathcal{M}_l : l \in [L]\}} \right) / 2}.$$

We can explicitly write down the ratio between the two desired probabilities as

$$\frac{\mathbb{P}_{\{\mathcal{M}_l \sim g(p_1), \mathcal{D}_l \sim \mathcal{M}_l : l \in [L]\}}(\{\mathcal{M}_l, \mathcal{D}_l : l \in [L]\})}{\mathbb{P}_{\{\mathcal{M}_l \sim g(p_2), \mathcal{D}_l \sim \mathcal{M}_l : l \in [L]\}}(\{\mathcal{M}_l, \mathcal{D}_l : l \in [L]\})} = \frac{(p_1)^\kappa (1 - p_1)^{L-\kappa}}{(p_2)^\kappa (1 - p_2)^{L-\kappa}}$$

where $\kappa = \sum_{l \in [L]} \mathbb{1}\{\mathbb{P}_{1,l}(s_g|s, a_g) = 1\}$.

As a result, with $p_h^\beta \in [\frac{1}{4}, \frac{3}{4}], \forall h \in [H]$ it holds that

$$\text{KL} \left( \mathbb{P}_{\{\mathcal{M}_l \sim g(p_1), \mathcal{D}_l \sim \mathcal{M}_l : l \in [L]\}} || \mathbb{P}_{\{\mathcal{M}_l \sim g(p_2), \mathcal{D}_l \sim \mathcal{M}_l : l \in [L]\}} \right)$$

$$= \mathbb{E}_{\{\mathcal{M}_l \sim g(p_1), \mathcal{D}_l \sim \mathcal{M}_l : l \in [L]\}} \left[ \kappa \log \left( \frac{p_1}{p_2} \right) + (L - \kappa) \log \left( \frac{1 - p_1}{1 - p_2} \right) \right]$$

$$= L p_1 \log \left( \frac{p_1}{p_2} \right) + (L - L p_1) \log \left( \frac{1 - p_1}{1 - p_2} \right)$$

$$\leq \frac{L(p_1 - p_2)^2}{p_2(1 - p_2)}$$

$$\leq 16 L (p_1 - p_2)^2$$

Finally, with $\delta = \frac{1}{16}\sqrt{\frac{2}{L}}$, it holds that

$$\mathbb{E}_{\{\mathcal{M}_l \sim g(p^\alpha), \mathcal{D}_l \sim \mathcal{M}_l : l \in [L]\}} \left[ \pi_h(a_b) \right] + \mathbb{E}_{\{\mathcal{M}_l \sim g(p^\beta), \mathcal{D}_l \sim \mathcal{M}_l : l \in [L]\}} \left[ 1 - \pi_h(a_g) \right]$$

$$\geq 1 - \sqrt{\text{KL} \left( \mathbb{P}_{\{\mathcal{M}_l \sim g(p^\alpha), \mathcal{D}_l \sim \mathcal{M}_l : l \in [L]\}} || \mathbb{P}_{\{\mathcal{M}_l \sim g(p^\beta), \mathcal{D}_l \sim \mathcal{M}_l : l \in [L]\}} \right) / 2}$$

$$\geq \frac{1}{2}$$

which concludes the proof as

$$\mathbb{E}_{\{\mathcal{M}_l \sim g(p_1), \mathcal{D}_l \sim \mathcal{M}_l : l \in [L]\}} \left[ \text{Gap}(\hat{\pi}; \mathcal{M}(p_1)) \right] + \mathbb{E}_{\{\mathcal{M}_l \sim g(p_2), \mathcal{D}_l \sim \mathcal{M}_l : l \in [L]\}} \left[ \text{Gap}(\hat{\pi}; \mathcal{M}(p_2)) \right]$$

$$\geq (H-1) \frac{1}{2} \cdot \frac{1}{16} \sqrt{\frac{2}{L}} = \Omega \left( \frac{H^2}{L} \right).$$

$\square$

## C    Upper Bound Analysis: Overview

In this section, we provide an overview with the steps shared in the proofs of each proposed algorithm. In other words, the following results are stated for all three proposed algorithms in general. The more challenging and specific proofs for our designs are illustrated in the following sections. The basic logistics here follow the seminal work of Jin et al. (2021b) in provably efficient offline RL.

The first step is to establish the validness of the pessimism. For different settings, various styles of pessimism are incorporated. Specifically, in this work, three penalty constructions are adopted in HetPEVI, HetPEVI-Adv and HetPEVI-Lin, respectively. As an abstraction, the validness of pessimism induced by penalties $\{\Gamma_h(s,a) : (s,a,h) \in \mathcal{S} \times \mathcal{A} \times [H]\}$ is defined in the following.

**Definition 2** (Validness of Pessimism). *For pessimistic value iterations using an estimated Bellman operator $\hat{\mathbb{B}}_h$ and penalties $\{\Gamma_h(s,a) : (s,a,h) \in \mathcal{S} \times \mathcal{A} \times [H]\}$, a valid pessimism is induced if the following event happens*

$$\mathcal{E} := \left\{ \left| \left( \hat{\mathbb{B}}_h \hat{V}_{h+1} \right)(s,a) - \left( \mathbb{B}_h \hat{V}_{h+1} \right)(s,a) \right| \leq \Gamma_h(s,a), \forall (s,a,h) \in \mathcal{S} \times \mathcal{A} \times [H] \right\}.$$

With a valid pessimism, we can obtain the following lemma.

**Lemma 5.** *Suppose that event $\mathcal{E}$ in Definition 2 happens, it holds that*

$$\zeta_h(s,a) := \left( \mathbb{B}_h \hat{V}_{h+1} \right)(s,a) - \hat{Q}_h(s,a) \in [0, 2\Gamma_h(s,a)], \qquad \forall (s,a,h) \in \mathcal{S} \times \mathcal{A} \times [H].$$

*Proof.* We can observe that for any $(s,a,h) \in \mathcal{S} \times \mathcal{A} \times [H]$, with the adopted pessimistic value iteration, it holds that

$$\begin{aligned}
\zeta_h(s,a) &= \left( \mathbb{B}_h \hat{V}_{h+1} \right)(s,a) - \hat{Q}_h(s,a) \\
&= \left( \mathbb{B}_h \hat{V}_{h+1} \right)(s,a) - \min\left\{ \left( \hat{\mathbb{B}}_h \hat{V}_{h+1} \right)(s,a) - \Gamma_h(s,a), H - h + 1 \right\}^+ \\
&\geq \min\left\{ \left( \mathbb{B}_h \hat{V}_{h+1} \right)(s,a) - \left( \hat{\mathbb{B}}_h \hat{V}_{h+1} \right)(s,a) + \Gamma_h(s,a), \left( \mathbb{B}_h \hat{V}_{h+1} \right)(s,a) \right\} \\
&\geq 0,
\end{aligned}$$

where the last inequality is due to event $\mathcal{E}$. Similar, for the other direction, it holds that

$$\begin{aligned}
\zeta_h(s,a) &= \left( \mathbb{B}_h \hat{V}_{h+1} \right)(s,a) - \hat{Q}_h(s,a) \\
&= \left( \mathbb{B}_h \hat{V}_{h+1} \right)(s,a) - \min\left\{ \left( \hat{\mathbb{B}}_h \hat{V}_{h+1} \right)(s,a) - \Gamma_h(s,a), H - h + 1 \right\}^+ \\
&\leq \max\left\{ \left( \mathbb{B}_h \hat{V}_{h+1} \right)(s,a) - \left( \hat{\mathbb{B}}_h \hat{V}_{h+1} \right)(s,a) + \Gamma_h(s,a), \left( \mathbb{B}_h \hat{V}_{h+1} \right)(s,a) - (H - h + 1) \right\} \\
&\leq 2\Gamma_h(s,a),
\end{aligned}$$

where the last inequality is due to event $\mathcal{E}$ and the fact that $(\mathbb{B}_h \hat{V}_{h+1})(s,a) \leq H - h + 1$. □

Furthermore, the suboptimality gap on the target MDP $\mathcal{M}$ between the output policy $\hat{\pi}$ and the optimal policy $\pi^*$ can be generally bounded via the following lemma.

**Lemma 6.** *Suppose that event $\mathcal{E}$ in Definition 2 happens, it holds that*

$$Gap(\hat{\pi}; \mathcal{M}) \leq 2 \sum_{h \in [H]} \mathbb{E}_{\pi^*, \mathcal{M}} \left[ \Gamma_h(s_h, a_h) \right].$$

*Proof.* It holds that

$$\text{Gap}(\hat{\pi}; \mathcal{M}) \overset{(a)}{=} - \sum_{h \in [H]} \mathbb{E}_{\hat{\pi}, \mathcal{M}} \left[ \zeta_h(s_h, a_h) \right] + \sum_{h \in [H]} \mathbb{E}_{\pi^*, \mathcal{M}} \left[ \zeta_h(s_h, a_h) \right]$$

$$+ \sum_{h \in [H]} \mathbb{E}_{\pi^*, \mathcal{M}} \left[ \hat{Q}_h(s_h, \pi^*(s_h)) - \hat{Q}_h(s_h, \hat{\pi}_h(s_h)) \right]$$

$$\overset{(b)}{\leq} 2 \sum_{h \in [H]} \mathbb{E}_{\pi^*, \mathcal{M}} \left[ \Gamma_h(s_h, a_h) \right],$$

where equation (a) is from Lemma 3.1 in (Jin et al., 2021b) (provided as Lemma 23), and inequality (b) is due to Lemma 5 together with the fact that $\hat{Q}_h(s_h, \cdot)$ takes maximum at $\hat{\pi}_h(s_h)$ in the proposed algorithms. □

In other words, $\text{Gap}(\hat{\pi}; \mathcal{M})$ is bounded via the sum of penalties $\Gamma_h(s_h, a_h)$ along the expected trajectory of optimal policy $\pi^*$ on the target MDP $\mathcal{M}$.

## D  THE HETPEVI ALGORITHM

### D.1  GOOD EVENT

**Lemma 7.** *The following event holds with probability at least $1 - \delta$ for HetPEVI*

$$\mathcal{G} := \Bigg\{ (i) \text{ The penalties } \{\Gamma_h(s, a)\} \text{ induce a valid pessimism;}$$

$$(ii) \ N_{h,l}(s, a) \geq cK d_{h,l}^{\rho_l}(s, a), \forall (l, s, a, h) \in [L] \times \mathcal{S} \times \mathcal{A} \times [H] \Bigg\}.$$

*Proof.* Part (I) is from Lemma 8 and part (II) obtained via Lemma 9. □

**Lemma 8.** *The penalty*

$$\Gamma_h(s, a) = c \sqrt{\sum_{l \in [L]} \frac{H^2}{(L^2 N_{h,l}(s, a)) \vee L}} + c \sqrt{\frac{H^2}{L}}$$

*in HetPEVI induces a valid pessimism with probability at least $1 - \delta$.*

*Proof.* For a fixed $(s, a, h)$, it holds that

$$\left( \hat{\mathbb{B}}_h \hat{V}_{h+1} \right)(s, a) - \left( \mathbb{B}_h \hat{V}_{h+1} \right)(s, a)$$

$$= \hat{r}_h(s, a) + \left( \hat{\mathbb{P}}_h \hat{V}_{h+1} \right)(s, a) - r_h(s, a) - \left( \mathbb{P}_h \hat{V}_{h+1} \right)(s, a)$$

$$= \sum_{l \in [L]} \frac{1}{L} \left( r_{h,l}(s, a) + \left( \hat{\mathbb{P}}_{h,l} \hat{V}_{h+1} \right)(s, a) \right) - \sum_{l \in [L]} \frac{1}{L} \left( r_{h,l}(s, a) + \left( \mathbb{P}_{h,l} \hat{V}_{h+1} \right)(s, a) \right)$$

$$+ \sum_{l \in [L]} \frac{1}{L} \left( r_{h,l}(s, a) + \left( \mathbb{P}_{h,l} \hat{V}_{h+1} \right)(s, a) \right) - \left( r_h(s, a) + \mathbb{P}_h \hat{V}_{h+1}(s, a) \right)$$

$$= \sum_{l \in [L]: N_{h,l}(s,a) > 0} \frac{1}{L} \left( r_{h,l}(s, a) + \sum_{k \in [K]: (s_{h,l}^k, a_{h,l}^k) = (s,a)} \frac{\hat{V}_{h+1}(s_{h+1,l}^k)}{N_{h,l}(s, a)} \right)$$

$$- \sum_{l \in [L]} \frac{1}{L} \left( r_{h,l}(s, a) + \left( \mathbb{P}_{h,l} \hat{V}_{h+1} \right)(s, a) \right)$$

$$+ \sum_{l \in [L]} \frac{1}{L} \left( r_{h,l}(s, a) + \left( \mathbb{P}_{h,l} \hat{V}_{h+1} \right)(s, a) \right) - \left( r_h(s, a) + \mathbb{P}_h \hat{V}_{h+1}(s, a) \right)$$

$$= \sum_{l \in [L]: N_{h,l}(s,a) > 0} \frac{1}{L} \left( \sum_{k \in [K]: (s_{h,l}^k, a_{h,l}^k) = (s,a)} \frac{\hat{V}_{h+1}(s_{h+1,l}^k)}{N_{h,l}(s, a)} - \left( \mathbb{P}_{h,l} \hat{V}_{h+1} \right)(s, a) \right)$$

$$+ \sum_{l\in[L]} \frac{1}{L} \left( r_{h,l}(s,a) + \left(\mathbb{P}_{h,l}\hat{V}_{h+1}\right)(s,a) \right) - \left( r_h(s,a) + \mathbb{P}_h\hat{V}_{h+1}(s,a) \right)$$

$$- \sum_{l\in[L]:N_{h,l}(s,a)=0} \frac{1}{L} \left( r_{h,l}(s,a) + \left(\mathbb{P}_{h,l}\hat{V}_{h+1}\right)(s,a) \right)$$

Then, via Hoeffding inequality, it can be recognized that with probability at least $1 - \delta$,

$$\left| \left(\hat{\mathbb{B}}_h\hat{V}_{h+1}\right)(s,a) - \left(\mathbb{B}_h\hat{V}_{h+1}\right)(s,a) \right|$$

$$\leq c\sqrt{\sum_{l\in[L]:N_{h,l}(s,a)>0} \frac{H^2}{L^2 N_{h,l}(s,a)}} + c\sqrt{\frac{H^2}{L}} + \sum_{l\in[L]:N_{h,l}(s,a)=0} \frac{H}{L}$$

$$= c\sqrt{\sum_{l\in[L]:N_{h,l}(s,a)>0} \frac{H^2}{L^2 N_{h,l}(s,a)} + \left(\sum_{l\in[L]:N_{h,l}(s,a)=0} \frac{H}{L}\right)^2} + c\sqrt{\frac{H^2}{L}}$$

$$= c\sqrt{\sum_{l\in[L]:N_{h,l}(s,a)>0} \frac{H^2}{L^2 N_{h,l}(s,a)} + \sum_{l\in[L]:N_{h,l}(s,a)=0} \frac{H^2}{L}} + c\sqrt{\frac{H^2}{L}}$$

$$= c\sqrt{\sum_{l\in[L]} \frac{H^2}{(L^2 N_{h,l}(s,a)) \vee L}} + c\sqrt{\frac{H^2}{L}}$$

$$\square$$

**Lemma 9.** *With probability at least $1 - \delta$, it holds that,*

$$N_{h,l}(s,a) \geq cKd_{h,l}^{\rho_l}(s,a), \quad \forall(l,s,a,h) \in [L] \times \mathcal{S} \times \mathcal{A} \times [H].$$

*Proof.* The proof can be done similarly as in Lemma B.3 of (Xie et al., 2021b). $\square$

## D.2 SUBOPTIMAITY GAP

*Proof of Theorems 2 and 3.* By Lemma 6 and 7, with probability at least $1 - \delta$, it holds that

$$\text{Gap}(\hat{\pi}; \mathcal{M}) \leq 2 \sum_{h\in[H]} \mathbb{E}_{\pi^*,\mathcal{M}}\left[\Gamma_h(s_h, a_h)\right].$$

With the individual coverage assumption (i.e., Assumption 2), it holds that

$$\sum_{h\in[H]} \sum_{(s,a)\in\mathcal{S}\times\mathcal{A}} d_h^{\pi^*}(s,a) \left[ \sqrt{\sum_{l\in[L]} \frac{H^2}{(L^2 N_{h,l}(s,a)) \vee L}} + \sqrt{\frac{H^2}{L}} \right]$$

$$\overset{(a)}{\leq} \sum_{h\in[H]} \sum_{(s,a)\in\mathcal{S}\times\mathcal{A}} d_h^{\pi^*}(s,a) \left[ \sqrt{\sum_{l\in[L]} \frac{H^2}{\left(L^2 K d_{h,l}^{\rho_l}(s,a)\right) \vee L}} + \sqrt{\frac{H^2}{L}} \right]$$

$$\overset{(b)}{=} \tilde{O}\left( \sqrt{\frac{C^* H^4 S}{LK}} + \sqrt{\frac{H^4}{L}} \right),$$

where inequality (a) is from (ii) of event $\mathcal{G}$ in Lemma 7, and equality (b) uses Assumption 2 and Cauchy-Schwarz inequality.

With the overall coverage assumption (i.e., Assumption 3), it can be similarly obtained that

$$\sum_{h\in[H]} \sum_{(s,a)\in\mathcal{S}\times\mathcal{A}} d_h^{\pi^*}(s,a) \left[ \sqrt{\sum_{l\in[L]} \frac{H^2}{(L^2 N_{h,l}(s,a)) \vee L}} + \sqrt{\frac{H^2}{L}} \right]$$

$$
\leq \sum_{h\in[H]} \sum_{(s,a)\in\mathcal{S}\times\mathcal{A}} d_h^{\pi^*}(s,a) \left[ \sqrt{\sum_{l\in[L]:d_{h,l}^{\rho_l}(s,a)>0} \frac{H^2}{L^2 K d_{h,l}^{\rho_l}(s,a)}} + \sqrt{\sum_{l\in[L]:d_{h,l}^{\rho_l}(s,a)=0} \frac{H^2}{L}} + \sqrt{\frac{H^2}{L}} \right]
$$

$$
\overset{(c)}{\leq} \sqrt{\frac{C^\dagger H^2}{LK}} \sum_{h\in[H]} \sum_{(s,a)\in\mathcal{S}\times\mathcal{A}} \sqrt{\mathbb{1}\{a=\pi_h^*(s)\} d_h^{\pi^*}(s,a)} + \sqrt{\frac{(L-L^\dagger)H^4}{L}} + \sqrt{\frac{H^4}{L}}
$$

$$
= \tilde{O}\left( \sqrt{\frac{C^\dagger H^4 S}{LK}} + \sqrt{\frac{(L+1-L^\dagger)H^4}{L}} \right),
$$

where inequality (c) is from Assumption 3. $\qquad\square$

## E   THE HETPEVI-ADV ALGORITHM

### E.1   ALGORITHM DESIGN

HetPEVI-Adv is designed as an enhanced version of HetPEVI with a Bernstein-type penalty term for the sample uncertainties. Especially, it shares the same procedure as HetPEVI except that the adopted penalty is

$$
\Gamma_h^\alpha(s,a) = c\sqrt{\sum_{l\in[L]} \frac{(\hat{\mathbb{V}}_{h,l}\hat{V}_{h+1})(s,a)}{L^2 N_{h,l}(s,a)}} + c\sqrt{\sum_{l\in[L]} \frac{H^2}{L^2 (N_{h,l}(s,a))^2}},
$$

where $(\hat{\mathbb{V}}_{h,l}\hat{V}_{h+1})(s,a)$ is the empirical variance of $\hat{V}_{h+1}(s')$ with $s' \sim \hat{\mathbb{P}}_{h,l}(\cdot|s,a)$. Note that compared with HetPEVI, the variance information is incorporated in HetPEVI-Adv.

### E.2   THEORETICAL ANALYSIS

**Theorem 6.** *Under Assumptions 1, 2, w.p. at least $1-\delta$, the output policy $\hat{\pi}$ of HetPEVI-Adv satisfies*

$$
Gap(\hat{\pi};\mathcal{M}) = \tilde{O}\left( \sqrt{\frac{C^* H^3 S}{LK}} + \frac{C^* H^3 S}{\sqrt{L}K} + \sqrt{\frac{C^* H^7 S}{L^2 K}} + \sqrt{\frac{H^4}{L}} \right). \tag{3}
$$

It is noted that the first three terms come from finite samples and the last term from finite sources. Furthermore, when $LK$ is sufficiently larger, the first term dominates the other second and third terms; thus, in this regime, it can be observed that HetPEVI-Adv has a tight performance loss to finite samples. However, the performance loss due to finite data sources still has an additional $H$ factor, which is left open for further investigations.

### E.3   GOOD EVENT

**Lemma 10.** *Under Assumptions 1, 2, with probability at least $1-\delta$, the following good event $\mathcal{G}$ happens, where*

$$
\mathcal{G} := \Big\{ (i) \text{ The penalties } \{\Gamma_h(s,a) : \forall(s,a,h)\in\mathcal{S}\times\mathcal{A}\times[H]\} \text{ induce a valid pessimism;}
$$

$$
(ii)\ V_h^*(s) - \hat{V}_h(s) = \tilde{O}\left( \sqrt{\frac{C^* H^4 S}{KL}} + \sqrt{\frac{H^4}{L}} \right), \forall(s,h)\in\mathcal{S}\times[H];
$$

$$
(iii)\ N_{h,l}(s,a) \geq cK d_{h,l}^{\rho_l}(s,a), \forall(l,s,a,h)\in[L]\times\mathcal{S}\times\mathcal{A}\times[H];
$$

$$
(iv)\ \left(\hat{\mathbb{V}}_{h,l}f\right)(s,a) \leq (\mathbb{V}_{h,l}f)(s,a) + c\sqrt{\frac{H^4}{N_{h,l}(s,a)}}, \forall(l,s,a,h)\in[L]\times\mathcal{S}\times\mathcal{A}\times[H];
$$

$$
(vii)\ \left(\bar{\mathbb{V}}_h f\right)(s,a) \leq (\mathbb{V}_h f)(s,a) + \sqrt{\frac{H^4}{L}}, \forall(s,a,h)\in\mathcal{S}\times\mathcal{A}\times[H] \Big\},
$$

*where $f$ is any fixed function: $\mathcal{S}\times\mathcal{A} \to [-H,H]$.*

*Proof.* Part (i) can be obtained similarly as Lemma 8 using the enhanced Bernstein inequality proved in Lemma 26, and the rest parts are established in the following lemmas. □

**Lemma 11.** *Under Assumptions 1, 2, with a probability of at least $1 - \delta$, the following crude bound of the output value of HetPEVI-Adv holds*

$$V_h^*(s) - \hat{V}_h(s) = \tilde{O}\left(\sqrt{\frac{C^* H^4 S}{KL}} + \sqrt{\frac{H^4}{L}}\right), \qquad \forall (s, a) \in \mathcal{S} \times [H]$$

*Proof.* It can be observed that

$$\Gamma_h^\alpha(s, a) = c\sqrt{\sum_{l \in [L]} \frac{(\hat{\mathbb{V}}_{h,l} \hat{V}_{h+1})(s, a)}{L^2 N_{h,l}(s, a)}} + c\sqrt{\sum_{l \in [L]} \frac{H^2}{L^2 (N_{h,l}(s, a))^2}} \le c\sqrt{\sum_{l \in [L]} \frac{H^2}{L^2 N_{h,l}(s, a)}}$$

Using the above upper bound of $\Gamma_h(s, a)$, the lemma can be obtained via similar steps of Theorem 2. □

**Lemma 12.** *With probability at least $1 - \delta$, the following events happen,*

$$N_{h,l}(s, a) \ge cK d_{h,l}^{\rho_l}(s, a), \qquad \forall (l, s, a, h) \in [L] \times \mathcal{S} \times \mathcal{A} \times [H]$$

*Proof.* The proof can be done similarly as in Lemma B.3 of (Xie et al., 2021b). □

**Lemma 13.** *With probability at least $1 - \delta$, for a fix functions $f : \mathcal{S} \times \mathcal{A} \to [-H, H]$, the following events happen*

$$\left(\hat{\mathbb{V}}_{h,l} f\right)(s, a) \le (\mathbb{V}_{h,l} f)(s, a) + c\sqrt{\frac{H^4}{N_{h,l}(s, a)}}, \qquad \forall (l, s, a, h) \in [L] \times \mathcal{S} \times \mathcal{A} \times [H].$$

*Proof.* With a union bound over $(s, a, h) \in \mathcal{S} \times \mathcal{A} \times [H]$, according to Hoeffding inequality, with probability at least $1 - \delta$, it holds that

$$\left(\hat{\mathbb{V}}_{h,l} f\right)(s, a) - (\mathbb{V}_{h,l} f)(s, a)$$
$$= \left[\left(\hat{\mathbb{P}}_{h,l} - \mathbb{P}_{h,l}\right) f^2\right](s, a) + \left[\left(\mathbb{P}_{h,l} - \hat{\mathbb{P}}_{h,l}\right) f\right](s, a) \cdot \left[\left(\mathbb{P}_{h,l} + \hat{\mathbb{P}}_{h,l}\right) f\right](s, a)$$
$$\le c\sqrt{\frac{H^4}{N_{h,l}(s, a)}},$$

which concludes the proof. □

**Lemma 14.** *With probability at least $1 - \delta$, for a fix functions $f : \mathcal{S} \times \mathcal{A} \to [-H, H]$, the following events happen*

$$\left(\bar{\mathbb{V}}_h f\right)(s, a) \le (\mathbb{V}_h f)(s, a) + c\sqrt{\frac{H^4}{L}}, \qquad \forall (s, a, h) \in \mathcal{S} \times \mathcal{A} \times [H].$$

*Proof.* With a union bound over $(s, a, h) \in \mathcal{S} \times \mathcal{A} \times [H]$, according to Hoeffding inequality, with probability at least $1 - \delta$, it holds that

$$\left(\bar{\mathbb{V}}_{h,l} f\right)(s, a) - (\mathbb{V}_{h,l} f)(s, a)$$
$$= \left[\left(\bar{\mathbb{P}}_h - \mathbb{P}_h\right) f^2\right](s, a) + \left[\left(\bar{\mathbb{P}}_h - \mathbb{P}_h\right) f\right](s, a) \cdot \left[\left(\bar{\mathbb{P}}_h + \mathbb{P}_h\right) f\right](s, a)$$
$$\le c\sqrt{\frac{H^4}{L}},$$

which concludes the proof. □

### E.4 SUBOPTIMAITY GAP

**Lemma 15.** *It holds that*

$$\sum_{h\in[H]}\sum_{(s,a)\in\mathcal{S}\times\mathcal{A}} d_h^{\pi^*}(s,a)[\mathbb{V}_h V_{h+1}^*](s,a) \le H^2.$$

*Proof.* The proof can be found in Lemma C.4 in Xie et al. (2021b). □

**Lemma 16.** *It holds that*

$$\sum_{l\in[L]} \left(\mathbb{V}_{h,l} V_{h+1}^*\right)(s,a) \le L\left(\bar{\mathbb{V}}_h V_{h+1}^*\right)(s,a).$$

*Proof.* This lemma is a direct consequence of Lemma 28. □

**Lemma 17.** *With event $\mathcal{G}$ happening, it holds that*

$$\sum_{h\in[H]}\sum_{(s,a)\in\mathcal{S}\times\mathcal{A}}\sum_{l\in[L]} d_h^{\pi^*}(s,a)\left(\mathbb{V}_{h,l}\hat{V}_{h+1}\right)(s,a) \le LH^2 + c\sqrt{\frac{C^*H^8 SL}{K}} + c\sqrt{H^8 L}.$$

*Proof.* First, the left-hand side can be decomposed as

$$\sum_{h\in[H]}\sum_{(s,a)\in\mathcal{S}\times\mathcal{A}}\sum_{l\in[L]} d_h^{\pi^*}(s,a)\left(\mathbb{V}_{h,l}\hat{V}_{h+1}\right)(s,a)$$

$$= \underbrace{\sum_{h\in[H]}\sum_{(s,a)\in\mathcal{S}\times\mathcal{A}} d_h^{\pi^*}(s,a) L\left(\mathbb{V}_h V_{h+1}^*\right)(s,a)}_{:=\text{term (I)}}$$

$$+ \underbrace{\sum_{h\in[H]}\sum_{(s,a)\in\mathcal{S}\times\mathcal{A}} d_h^{\pi^*}(s,a) L\left(\left(\bar{\mathbb{V}}_h V_{h+1}^*\right)(s,a) - \left(\mathbb{V}_h V_{h+1}^*\right)(s,a)\right)}_{:=\text{term (II)}}$$

$$+ \underbrace{\sum_{h\in[H]}\sum_{(s,a)\in\mathcal{S}\times\mathcal{A}} d_h^{\pi^*}(s,a)\left(\sum_{l\in[L]}\left(\mathbb{V}_{h,l} V_{h+1}^*\right)(s,a) - L\left(\bar{\mathbb{V}}_h V_{h+1}^*\right)(s,a)\right)}_{:=\text{term (III)}}$$

$$+ \underbrace{\sum_{h\in[H]}\sum_{(s,a)\in\mathcal{S}\times\mathcal{A}}\sum_{l\in[L]} d_h^{\pi^*}(s,a)\left[\left(\mathbb{V}_{h,l}\hat{V}_{h+1}\right)(s,a) - \left(\mathbb{V}_{h,l} V_{h+1}^*\right)(s,a)\right]}_{:=\text{term (IV)}}$$

Then, for term (I), with Lemma 15, it holds that

$$\text{term (I)} \le LH^2.$$

For term (II), with event (vii) in Lemma 10 it holds that

$$\text{term (II)} = \sum_{h\in[H]}\sum_{(s,a)\in\mathcal{S}\times\mathcal{A}} d_h^{\pi^*}(s,a) L\left(\left(\bar{\mathbb{V}}_h V_{h+1}^*\right)(s,a) - \left(\mathbb{V}_h V_{h+1}^*\right)(s,a)\right)$$

$$\le c\sum_{h\in[H]}\sum_{(s,a)\in\mathcal{S}\times\mathcal{A}} d_h^{\pi^*}(s,a) L\sqrt{\frac{H^4}{L}}$$

$$= c\sqrt{H^6 L}$$

For term (III), with Lemma 16, it holds that

$$\text{term (III)} \le 0.$$

For term (IV), we can obtain that

$$\text{term (IV)} = \sum_{h\in[H]} \sum_{(s,a)\in\mathcal{S}\times\mathcal{A}} \sum_{l\in[L]} d_h^{\pi^*}(s,a) \left[ \left(\mathbb{V}_{h,l}\hat{V}_{h+1}\right)(s,a) - \left(\mathbb{V}_{h,l}V_{h+1}^*\right)(s,a) \right]$$

$$\overset{(a)}{\leq} 4H \sum_{h\in[H]} \sum_{(s,a)\in\mathcal{S}\times\mathcal{A}} \sum_{l\in[L]} d_h^{\pi^*}(s,a) \left\| \hat{V}_{h+1}(\cdot) - V_{h+1}^*(\cdot) \right\|_\infty$$

$$\overset{(b)}{\leq} cH \sum_{h\in[H]} \sum_{(s,a)\in\mathcal{S}\times\mathcal{A}} \sum_{l\in[L]} d_h^{\pi^*}(s,a) \cdot \left( \sqrt{\frac{C^*H^4S}{LK}} + \sqrt{\frac{H^4}{L}} \right)$$

$$= c\sqrt{\frac{C^*H^8SL}{K}} + c\sqrt{H^8L},$$

where inequality (a) is from simple algebraic based on the definition of variance and inequality (b) is from part (ii) of even $\mathcal{G}$ in Lemma 10. $\quad\square$

*Proof of Theorem 6.* By Lemma 6 and 10, with probability at least $1-\delta$, it holds that

$$\text{Gap}(\hat{\pi}; \mathcal{M}) \leq 2 \sum_{h\in[H]} \mathbb{E}_{\pi^*, \mathcal{M}} \left[ \Gamma_h(s_h, a_h) \right].$$

Furthermore, it can be obtained that

$$\Gamma_h^\alpha(s_h, a_h) = c\sqrt{\sum_{l\in[L]} \frac{\left(\hat{\mathbb{V}}_{h,l}\hat{V}_{h+1}\right)(s,a)}{L^2 N_{h,l}(s,a)}} + c\sqrt{\sum_{l\in[L]} \frac{H^2}{L^2(N_{h,l}(s,a))^2}}$$

$$\overset{(a)}{\leq} c\sqrt{\sum_{l\in[L]} \frac{\left(\mathbb{V}_{h,l}\hat{V}_{h+1}\right)(s,a)}{L^2 N_{h,l}(s,a)}} + c\sqrt{\sum_{l\in[L]} \frac{H^2}{L^2(N_{h,l}(s,a))^{3/2}}} + \sqrt{\sum_{l\in[L]} \frac{H^2}{L^2(N_{h,l}(s,a))^2}}$$

$$\leq c\sqrt{\sum_{l\in[L]} \frac{\left(\mathbb{V}_{h,l}\hat{V}_{h+1}^{\text{ref}}\right)(s,a)}{L^2 N_{h,l}(s,a)}} + \sqrt{\sum_{l\in[L]} \frac{1}{L^2 N_{h,l}(s,a)}} + \sqrt{\sum_{l\in[L]} \frac{H^4}{L^2(N_{h,l}(s,a))^2}},$$

where inequality (a) is from event (iv) of Lemma 10. Then, for each separate term, it holds that

$$\text{term (I)} := c \sum_{h\in[H]} \sum_{(s,a)\in\mathcal{S}\times\mathcal{A}} d_h^{\pi^*}(s,a) \sqrt{\sum_{l\in[L]} \frac{\left(\mathbb{V}_{h,l}\hat{V}\right)(s,a)}{L^2 N_{h,l}(s,a)}}$$

$$\overset{(a)}{\leq} c \sum_{h\in[H]} \sum_{(s,a)\in\mathcal{S}\times\mathcal{A}} d_h^{\pi^*}(s,a) \sqrt{\sum_{l\in[L]} \frac{\left(\mathbb{V}_{h,l}\hat{V}_{h+1}\right)(s,a)}{L^2 K d_{h,l}^{\rho_l}(s,a)}}$$

$$\leq c\sqrt{\frac{C^*HS}{L^2K}} \sqrt{\sum_{h\in[H]} \sum_{(s,a)\in\mathcal{S}\times\mathcal{A}} \sum_{l\in[L]} d_h^{\pi^*}(s,a)\mathbb{1}\{a=\pi_h^*(s)\} \left(\mathbb{V}_{h,l}\hat{V}_{h+1}\right)(s,a)}$$

$$\overset{(b)}{\leq} c\sqrt{\frac{C^*HS}{L^2K}} \sqrt{LH^2 + \sqrt{\frac{C^*H^8SL}{K}} + \sqrt{H^8L}}$$

$$\leq c\sqrt{\frac{C^*HS}{L^2K}} \sqrt{LH^2 + \frac{C^*H^2SL}{K} + H^6}$$

$$\leq c\sqrt{\frac{C^*H^3S}{LK}} + c\frac{C^*H^2S}{\sqrt{L}K} + c\sqrt{\frac{C^*H^7S}{L^2K}};$$

$$\text{term (II)} := c \sum_{h\in[H]} \sum_{(s,a)\in\mathcal{S}\times\mathcal{A}} d_h^{\pi^*}(s,a) \sqrt{\sum_{l\in[L]} \frac{1}{L^2 N_{h,l}(s,a)}}$$

$$\overset{(c)}{\leq} c \sum_{h \in [H]} \sum_{(s,a) \in \mathcal{S} \times \mathcal{A}} d_h^{\pi^*}(s,a) \sqrt{\sum_{l \in [L]} \frac{1}{KL^2 d_{h,l}^{\rho_l}(s,a)}}$$

$$\leq c \sqrt{\frac{C^*}{KL}} \sum_{h \in [H]} \sum_{(s,a) \in \mathcal{S} \times \mathcal{A}} \sqrt{d_h^{\pi^*}(s,a) \mathbb{1}\{a = \pi_h^*(s)\}}$$

$$\leq c \sqrt{\frac{C^* H^2 S}{LK}};$$

$$\text{term (I.c)} := c \sum_{h \in [H]} \sum_{(s,a) \in \mathcal{S} \times \mathcal{A}} d_h^{\pi^*}(s,a) \sqrt{\sum_{l \in [L]} \frac{H^4}{L^2 (N_{h,l}(s,a))^2}}$$

$$\overset{(d)}{\leq} c \sum_{h \in [H]} \sum_{(s,a) \in \mathcal{S} \times \mathcal{A}} d_h^{\pi^*}(s,a) \sqrt{\sum_{l \in [L]} \frac{H^4}{L^2 (K d_{h,l}^{\rho_l}(s,a))^2}}$$

$$\leq c \sum_{h \in [H]} \sum_{(s,a) \in \mathcal{S} \times \mathcal{A}} \mathbb{1}\{\pi_h^*(s) = a\} \sqrt{\sum_{l \in [L]} \frac{(C^*)^2 H^4}{L^2 K^2}}$$

$$\leq c \frac{C^* S H^3}{\sqrt{L} K},$$

where inequalities (a), (c) and (d) are from event (iv) of Lemma 10 and inequality (b) is from Lemma 17.

By aggregating these three terms together and adding the sum of source uncertainties, it can be observed that

$$\text{Gap}(\hat{\pi}; \mathcal{M}) := \tilde{O}\left( \sqrt{\frac{C^* H^3 S}{LK}} + \frac{C^* H^3 S}{\sqrt{L} K} + \sqrt{\frac{C^* H^7 S}{L^2 K}} + \sqrt{\frac{H^4}{L}} \right).$$

$\square$

# F    THE HETPEVI-LIN ALGORITHM

## F.1    PROPERTIES OF LINEAR MDPS

*Proof of Lemma 1.* This proof is standard for studies in linear MDPs (Jin et al., 2020; 2021b). We include it here for completeness and to facilitate the analysis of the next lemma. Based on the Bellman equation, it holds that

$$(\mathbb{B}_{h,l} f)(s,a) = r_{h,l}(s,a) + (\mathbb{P}_{h,l} f)(s,a) = \langle \phi(s,a), \theta_{h,l} \rangle + \int_{\mathcal{S}} f(s') \cdot \langle \phi(s,a), d\mu_h(s') \rangle,$$

where means that $(\mathbb{B}_{h,l} f)(s,a) = \langle \phi(s,a), w_{h,l}^f \rangle$ with $w_{h,l}^f = \theta_{h,l} + \int_{\mathcal{S}} f(s') \, d\mu_{h,l}(s')$. Similar arguments hold for $(\mathbb{B}_h f)(s,a) = \langle \phi(s,a), w_h^f \rangle$ with $w_h^f = \theta_h + \int_{\mathcal{S}} f(s') \, d\mu_h(s')$. $\square$

*Proof of Lemma 2.* First, it can be observed that the transition and reward of the sample average MDP $\bar{\mathcal{M}}$ is linear since

$$\bar{\mathbb{P}}_h(s'|s,a) = \sum_{l \in [L]} \mathbb{P}_{h,l}(s'|s,a)/L = \sum_{l \in [L]} \langle \phi(s,a), \mu_{h,l}(s') \rangle / L = \langle \phi(s,a), \bar{\mu}_h(s') \rangle;$$

$$\bar{r}_h(s,a) = \sum_{l \in [L]} r_{h,l}(s,a)/L = \sum_{l \in [L]} \langle \phi(s,a), \theta_{h,l} \rangle / L = \langle \phi(s,a), \bar{\theta}_h \rangle,$$

where

$$\bar{\mu}_h(s') := \sum_{l \in [L]} \mu_{h,l}(s')/L; \qquad \bar{\theta}_h := \sum_{l \in [L]} \theta_{h,l}/L.$$

Then, we can verify the constraints in Definition 1 as

$$\|\phi(s,a)\|_2 \le 1, \qquad \forall (s,a) \in \mathcal{S} \times \mathcal{A},$$

and

$$
\begin{aligned}
\max\{\|\bar{\mu}_h(\mathcal{S})\|_2, \|\bar{\theta}_h\|_2\} &= \max \left\{ \left\| \sum_{l \in [L]} \mu_{h,l}(\mathcal{S})/L \right\|_2, \left\| \sum_{l \in [L]} \theta_{h,l}/L \right\|_2 \right\} \\
&\le \max \left\{ \sum_{l \in [L]} \|\mu_{h,l}(\mathcal{S})\|_2 / L, \sum_{l \in [L]} \|\theta_{h,l}\|_2 / L \right\} \\
&\le \sqrt{d}, \qquad \forall h \in [H].
\end{aligned}
$$

Moreover, it holds that

$$
\begin{aligned}
(\bar{\mathbb{B}}_h f)(s,a) &= \bar{r}_h(s,a) + (\bar{\mathbb{P}}_h f)(s,a) \\
&= \langle \phi(s,a), \bar{\theta}_h \rangle + \int_{\mathcal{S}} f(s') \cdot \langle \phi(s,a), \mathrm{d}\bar{\mu}_h(s') \rangle \\
&= \left\langle \phi(s,a), \sum_{l \in [L]} \theta_{h,l}/L \right\rangle + \int_{\mathcal{S}} f(s') \cdot \left\langle \phi(s,a), \sum_{l \in [L]} \mathrm{d}\mu_{h,l}(s')/L \right\rangle \\
&= \left\langle \phi(s,a), \sum_{l \in [L]} \left( \theta_{h,l} + \int_{\mathcal{S}} f(s')\, \mathrm{d}\mu_{h,l}(s') \right) /L \right\rangle \\
&= \left\langle \phi(s,a), \sum_{l \in [L]} w_{h,l}^f / L \right\rangle \\
&= \left\langle \phi(s,a), \bar{w}_h^f \right\rangle,
\end{aligned}
$$

which proves the claim. $\qquad\square$

### F.2 GOOD EVENT

**Lemma 18.** *The following event holds with probability at least $1 - \delta$ for HetPEVI-Lin*

$$\mathcal{G} := \left\{ (i) \text{ The penalties } \{\Gamma_h(s,a)\} \text{ induce a valid pessimism}; \right.$$

$$\left. (ii) \left\| \Xi_{h,l} - \hat{\Xi}_{h,l} \right\|_2 \le c\sqrt{\frac{1}{K}}, \forall (l,h) \in [L] \times [H] \right\},$$

*where*

$$
\begin{aligned}
\Xi_{h,l} &:= \mathbb{E}_{\rho_l, \mathcal{M}_l} \left[ \phi(s_h, a_h)\phi(s_h, a_h)^\top \right] \\
\hat{\Xi}_{h,l} &:= \frac{1}{K} \sum_{k \in [K]} \phi(s_{h,l}^k, a_{h,l}^k)\phi(s_{h,l}^k, a_{h,l}^k)^\top.
\end{aligned}
$$

*Proof.* The part (i) holds according to Lemma 22 and the part (ii) can be observed via the Lemma 30. $\qquad\square$

In the following, for all $h \in [H]$, based on the assumption that the matrix $\sum_{l \in [L]} \Lambda_{h,l}^{-1}$ is invertible, we denote that

$$\Upsilon_h := \left( \sum_{l \in [L]} \Lambda_{h,l}^{-1} \right)^{-1}; \qquad \Upsilon_h^{-1} := \sum_{l \in [L]} \Lambda_{h,l}^{-1}.$$

**Lemma 19.** *With the penalties* $\Gamma_h^\alpha(s,a) = cH\left(\sqrt{\frac{d}{L^2}} + \sqrt{\frac{d\lambda}{L}}\right)\|\phi(s,a)\|_{\Upsilon_h^{-1}}$ *in HetPEVI-Lin,* *with probability at least* $1 - \delta$, *it holds that for all* $(s,a,h) \in \mathcal{S} \times \mathcal{A} \times [H]$

$$\left|\left(\hat{\mathbb{B}}_h \hat{V}_{h+1}\right)(s,a) - \left(\bar{\mathbb{B}}_h \hat{V}_{h+1}\right)(s,a)\right| \le \Gamma_h^\alpha(s,a).$$

*Proof.* For a fixed $h$ and a fixed function $\hat{V}_{h+1}(\cdot) : \mathcal{S} \to \mathbb{R}$, with Lemma 2, there exists $w_h \in \mathbb{R}^d$ such that

$$\left(\bar{\mathbb{B}}_h \hat{V}_{h+1}\right)(s,a) = \langle \phi(s,a), \bar{w}_h \rangle, \qquad \forall (s,a) \in \mathcal{S} \times \mathcal{A}.$$

With $(\hat{\mathbb{B}}_h \hat{V}_{h+1})(s,a) := \langle \phi(s,a), \hat{w}_h \rangle$, it holds that

$$\left|\left(\bar{\mathbb{B}}_h \hat{V}_{h+1}\right)(s,a) - \left(\hat{\mathbb{B}}_h \hat{V}_{h+1}\right)(s,a)\right| = |\langle \phi(s,a), \bar{w}_h - \hat{w}_h \rangle| \le \|\phi(s,a)\|_{\Upsilon_h^{-1}} \|\bar{w}_h - \hat{w}_h\|_{\Upsilon_h},$$

and thus we can instead control $\|\bar{w}_h - \hat{w}_h\|_{\Upsilon_h}$. For this purpose, we can observe that

$$\|\bar{w}_h - \hat{w}_h\|_{\Upsilon_h} = \langle \bar{w}_h - \hat{w}_h, (\Upsilon_h)^{1/2} X \rangle,$$

where

$$X := \frac{(\Upsilon_h)^{1/2}(\bar{w}_h - \hat{w}_h)}{\|\bar{w}_h - \hat{w}_h\|_{\Upsilon_h}} \in \mathbb{S}^{d-1}.$$

With Lemma 29, we can find a $\varepsilon$-covering $\mathcal{C}_\varepsilon$ over $\mathbb{S}^{d-1}$ with $|\mathcal{C}_\varepsilon| \le (3/\varepsilon)^d$. Using Lemma 20 and a union bound, we can have that with probability $1 - \delta$, $\forall (y,h) \in \mathcal{C}_\varepsilon \times [H]$, it holds that

$$\left|\langle (\Upsilon_h)^{1/2} y, \bar{w}_h - \hat{w}_h \rangle\right| \le cH\left(\sqrt{\frac{\log(H|\mathcal{C}_\varepsilon|/\delta)}{L^2}} + \sqrt{\frac{d\lambda}{L}}\right) \left\|(\Upsilon_h)^{1/2} y\right\|_{(\Upsilon_h)^{-1}}$$

$$\le cH\left(\sqrt{\frac{d\log(H/(\varepsilon\delta))}{L^2}} + \sqrt{\frac{d\lambda}{L}}\right).$$

With this event happening, for any $x \in \mathbb{S}^{d-1}$, there exists $y \in \mathcal{C}_\varepsilon$ such that $\|x - y\|_2 \le \varepsilon$ and further it holds that

$$\|\bar{w}_h - \hat{w}_h\|_{\Upsilon_h} = \max_{x \in \mathbb{S}^{d-1}} \langle \bar{w}_h - \hat{w}_h, (\Upsilon_h)^{1/2} x \rangle$$

$$= \max_{x \in \mathbb{S}^{d-1}} \min_{y \in \mathcal{C}_\varepsilon} \left[\langle \bar{w}_h - \hat{w}_h, (\Upsilon_h)^{1/2} y \rangle + \langle \bar{w}_h - \hat{w}_h, (\Upsilon_h)^{1/2}(x - y) \rangle\right]$$

$$\le \max_{x \in \mathbb{S}^{d-1}} \min_{y \in \mathcal{C}_\varepsilon} \left[cH\left(\sqrt{\frac{d\log(H/(\varepsilon\delta))}{L^2}} + \sqrt{\frac{d\lambda}{L}}\right) + \|\bar{w}_h - \hat{w}_h\|_{\Upsilon_h} \|x - y\|_2\right]$$

$$\le cH\left(\sqrt{\frac{d\log(H/(\varepsilon\delta))}{L^2}} + \sqrt{\frac{d\lambda}{L}}\right) + \varepsilon \|\bar{w}_h - \hat{w}_h\|_{\Upsilon_h}.$$

Thus, with $\varepsilon = 1/2$, with probability at least $1 - \delta$, $\forall h \in [H]$, it holds that

$$\|w_h - \hat{w}_h\|_{\Upsilon_h} \le cH\left(\sqrt{\frac{d}{L^2}} + \sqrt{\frac{d\lambda}{L}}\right).$$

Thus, for any $(s,a) \in \mathcal{S} \times \mathcal{A}$, it holds that

$$\left|\left(\bar{\mathbb{B}}_h \hat{V}_{h+1}\right)(s,a) - \left(\hat{\mathbb{B}}_h \hat{V}_{h+1}\right)(s,a)\right| \le \|\phi(s,a)\|_{\Upsilon_h^{-1}} \|w_h - \hat{w}_h\|_{\Upsilon_h}$$

$$\le cH\left(\sqrt{\frac{d}{L^2}} + \sqrt{\frac{d\lambda}{L}}\right) \|\phi(s,a)\|_{\Upsilon_h^{-1}}.$$

With a union bound over $h \in [H]$, the lemma is proved. $\qquad\square$

**Lemma 20.** *For a fixed vector $x \in \mathbb{R}^d$ and a fixed $h \in [H]$, with probability at least $1 - \delta$, it holds that*

$$|\langle x, \bar{w}_h - \hat{w}_h \rangle| \leq \left( \sqrt{\frac{2 \log(2/\delta)}{L^2}} + \sqrt{\frac{dH^2 \lambda}{L}} \right) \|x\|_{\Upsilon_h^{-1}}.$$

*Proof.* It holds that

$$x^\top (\bar{w}_h - \hat{w}_h) = x^\top \left( \sum_{l \in [L]} w_{h,l}/L - \sum_{l \in [L]} \hat{w}_{h,l}/L \right)$$

$$\overset{(a)}{=} x^\top \sum_{l \in [L]} \frac{1}{L} \Lambda_{h,l}^{-1} \left( \sum_{k \in [K]} \phi(s_{h,l}^k, a_{h,l}^k) \phi(s_{h,l}^k, a_{h,l}^k)^\top + \lambda I \right) w_{h,l}$$

$$- x^\top \sum_{l \in [L]} \frac{1}{L} \Lambda_{h,l}^{-1} \left( \sum_{k \in [K]} \phi(s_{h,l}^k, a_{h,l}^k) \left( r_{h,l}^k + \hat{V}_{h+1}(s_{h+1,l}^k) \right) \right)$$

$$= \underbrace{\lambda x^\top \sum_{l \in [L]} \frac{1}{L} \Lambda_{h,l}^{-1} w_{h,l}}_{\text{term (I)}}$$

$$+ \underbrace{x^\top \sum_{l \in [L]} \frac{1}{L} \Lambda_{h,l}^{-1} \left( \sum_{k \in [K]} \phi(s_{h,l}^k, a_{h,l}^k) \left( \phi(s_{h,l}^k, a_{h,l}^k)^\top w_{h,l} - \left( r_{h,l}^k + \hat{V}_{h+1}(s_{h+1,l}^k) \right) \right) \right)}_{\text{term (II)}},$$

where the definition of $\Lambda_{h,l}$ and $\hat{w}_{h,l}$ is used in equation (a).

For term (I), we have

$$\lambda \left| x^\top \sum_{l \in [L]} \frac{1}{L} \Lambda_{h,l}^{-1} w_{h,l} \right| = \lambda \left| \sum_{l \in [L]} x^\top \Lambda_{h,l}^{-1} w_{h,l}/L \right| \leq \frac{\lambda}{L} \|x\|_{\Upsilon_h^{-1}} \sqrt{\sum_{l \in [L]} w_{h,l}^\top \Lambda_{h,l}^{-1} w_{h,l}}$$

$$\overset{(a)}{\leq} \frac{\lambda}{L} \|x\|_{\Upsilon_h^{-1}} \sqrt{4H^2 L d/\lambda} = c \sqrt{\frac{dH^2 \lambda}{L}} \|x\|_{\Upsilon_h^{-1}},$$

where inequality (a) is based on the fact that $\|w_{h,l}\|_2 \leq 2H\sqrt{d}, \forall (l, h) \in [L] \times [H]$ (which can be proved similarly to Lemma B.1 in Jin et al. (2020)) and the following observation:

$$w_{h,l}^\top \Lambda_{h,l}^{-1} w_{h,l} \leq \|w_h\|_2^2 \|\Lambda_{h,l}^{-1}\|_2 \leq 4dH^2/\lambda.$$

For term (II), with

$$t := \frac{H}{L} \sqrt{2 x^\top \Upsilon_h^{-1} x \log(2/\delta)},$$

conditioned on the state-action pairs at step $h$, we have

$$\mathbf{P} \left( \left| x^\top \sum_{l \in [L]} \frac{1}{L} \Lambda_{h,l}^{-1} \left( \sum_{k \in [K]} \phi(s_{h,l}^k, a_{h,l}^k) \left( \phi(s_{h,l}^k, a_{h,l}^k)^\top w_{h,l} - \left( r_{h,l}^k + \hat{V}_{h+1}(s_{h+1,l}^k) \right) \right) \right) \right| \geq t \right)$$

$$\leq 2 \exp \left( -\frac{2t^2}{4H^2 \sum_{l \in [L]} \sum_{k \in [K]} \frac{1}{L^2} \left( x^\top \Lambda_{h,l}^{-1} \phi(s_{h,l}^k, a_{h,l}^k) \right)^2} \right)$$

$$\leq 2 \exp \left( -\frac{x^\top \Upsilon_h^{-1} x \log(2/\delta)}{\sum_{l \in [L]} \sum_{k \in [K]} x^\top \Lambda_{h,l}^{-1} \phi(s_{h,l}^k, a_{h,l}^k) \phi(s_{h,l}^k, a_{h,l}^k)^\top \Lambda_{h,l}^{-1} x} \right)$$

$$\leq 2\exp\left(-\frac{x^\top \Upsilon_h^{-1} x \log(2/\delta)}{\sum_{l\in[L]}\sum_{k\in[K]} x^\top \Lambda_{h,l}^{-1}\left(\phi(s_{h,l}^k,a_{h,l}^k)\phi(s_{h,l}^k,a_{h,l}^k)^\top + \lambda I\right)\Lambda_{h,l}^{-1} x}\right)$$

$$\leq 2\exp\left(-\frac{x^\top \Upsilon_h^{-1} x \log(2/\delta)}{x^\top \Upsilon_h^{-1} x}\right)$$

$$= \delta,$$

which leads to the claim. $\qquad\square$

**Lemma 21.** *For the penalties $\Gamma_h^\beta(s,a) = c\sqrt{\frac{dH^2}{L}}$ in HetPEVI, with probability at least $1-\delta$, it holds that for all $(s,a,h) \in \mathcal{S}\times\mathcal{A}\times[H]$,*

$$\left|\left(\bar{\mathbb{B}}_h \hat{V}_{h+1}\right)(s,a) - \left(\mathbb{B}_h \hat{V}_{h+1}\right)(s,a)\right| \leq \Gamma_h^\beta(s,a).$$

*Proof.* For a fixed $\phi(s,a) \in \mathbb{R}^d$ and a fixed $\hat{V}_{h+1}$, since $(\mathbb{B}_{h,l}\hat{V}_{h+1})(s,a)$ is bounded between $[0,H]$ and has an expectation of $(\mathbb{B}_h \hat{V}_{h+1})(s,a)$, it is then a $H$-sub-Gaussian random variable (Vershynin, 2018). Thus, it holds that

$$\mathbf{P}\left(\left|\left(\bar{\mathbb{B}}_h \hat{V}_{h+1}\right)(s,a) - \left(\mathbb{B}_h \hat{V}_{h+1}\right)(s,a)\right| \geq c\sqrt{\frac{H^2 \log(2H/\delta)}{L}}\right) \leq \delta, \qquad \forall h \in [H].$$

Similarly as in Lemma 19, using a covering argument, it holds that

$$\mathbf{P}\left(\left|\left(\bar{\mathbb{B}}_h \hat{V}_{h+1}\right)(s,a) - \left(\mathbb{B}_h \hat{V}_{h+1}\right)(s,a)\right| \geq c\sqrt{\frac{dH^2}{L}}\right) \leq \delta, \qquad \forall (s,a,h) \in \mathcal{S}\times\mathcal{A}\times[H],$$

which concludes the proof. $\qquad\square$

**Lemma 22.** *The penalties $\{\Gamma_h(s,a) = \Gamma_h^\alpha(s,a) + \Gamma_h^\beta(s,a) : (s,a,h) : (s,a,h) \in \mathcal{S}\times\mathcal{A}\times[H]\}$ in HetPEVI induce a valid pessimism with probability at least $1-\delta$ with respect to the estimated Bellman operator $\hat{\mathbb{B}}_h$ as $(\hat{\mathbb{B}}_h \hat{V}_{h+1})(s,a) := \langle \phi(s,a), \hat{w}_h\rangle$.*

*Proof.* This result can be obtained by combining Lemmas 19 and 21. $\qquad\square$

### F.3 SUBOPTIMALITY GAP

*Proof of Theorems 4.* By Lemma 6 and 18, with probability at least $1-\delta$, it holds that

$$\mathrm{Gap}(\hat{\pi};\mathcal{M}) \leq 2\sum_{h\in[H]} \mathbb{E}_{\pi^*,\mathcal{M}}\left[\Gamma_h(s_h,a_h)\right].$$

We denote

$$\Sigma_h = \mathbb{E}_{\pi^*,\mathcal{M}}\left[\phi(s_h,a_h)\phi(s_h,a_h)^\top\right],$$

which is positive semi-definite as $\Xi_{h,l}$ and $\hat{\Xi}_{h,l}$ for all $h \in [H]$. Correspondingly, we denote $\mathrm{Rank}_h = \mathrm{Rank}(\Sigma_h)$ for all $h \in [H]$. Also, for a positive semi-definite matrix $\Sigma \in \mathbb{R}^{d\times d}$, we denote its ordered eigenvalues as

$$\gamma_1(\Sigma) \geq \gamma_1(\Sigma) \cdots \geq \gamma_d(\Sigma) \geq 0.$$

With Assumption 5, when $K \geq c\max_{h,l}\left\{\frac{(D^*)^2}{(\gamma_{\mathrm{Rank}_h}(\Sigma_h))^2}\right\}$ and $\lambda = 1/L$, it holds that

$$\sum_{h\in[H]} \mathbb{E}_{\pi^*,\mathcal{M}}\left[\sqrt{\frac{dH^2}{L^2}}\|\phi(s_h,a_h)\|_{\Upsilon_h^{-1}} + \sqrt{\frac{dH^2}{L}}\right]$$

$$\overset{(a)}{\leq} \sqrt{\frac{dH^2}{L^2}}\sum_{h\in[H]}\sqrt{\sum_{l\in[L]}\mathrm{Tr}\left(\mathbb{E}_{\pi^*,\mathcal{M}}[\phi(s_h,a_h)\phi(s_h,a_h)^\top]\Lambda_{h,l}^{-1}\right)} + \sqrt{\frac{dH^4}{L}}$$

$$\overset{(b)}{\leq} c\sqrt{\frac{dH^2}{L^2}} \sum_{h\in[H]} \sqrt{\sum_{l\in[L]}\sum_{i\in[d]} \frac{\gamma_i(\Sigma_h)}{\gamma_i(\Lambda_{h,l})}} + \sqrt{\frac{dH^4}{L}}$$

$$\overset{(c)}{\leq} \sqrt{\frac{dH^2}{L^2}} \sum_{h\in[H]} \sqrt{\sum_{l\in[L]}\sum_{i\in[\text{Rank}_l]} \frac{\gamma_i(\Sigma_l)/K}{1/(KL) + \gamma_i(\hat{\Xi}_{h,l})}} + \sqrt{\frac{dH^4}{L}}$$

$$\overset{(d)}{\leq} c\sqrt{\frac{dH^2}{L^2}} \sum_{h\in[H]} \sqrt{\sum_{l\in[L]}\sum_{i\in[\text{Rank}_h]} \frac{\gamma_i(\Sigma_h)/K}{1/(KL) + \gamma_i(\Sigma_l)/D^*}} + \sqrt{\frac{dH^4}{L}}$$

$$\leq c\sqrt{\frac{d^2H^4D^*}{LK}} + \sqrt{\frac{dH^4}{L}}$$

where inequality (a) is from the Cauchy-Schwarz inequality, inequality (b) is from the Von Neumann's trace inequality (Mirsky, 1975) and removes the zero eigenvalues of $\Sigma_h$ from the sum. Furthermore, for $i \leq \text{Rank}_h$ (implying $\gamma_i(\Sigma_h) > 0$), inequality (d) is from the following observation

$$\gamma_i(\hat{\Xi}_{h,l}) \overset{(e)}{\geq} \gamma_i(\Xi_{h,l}) - \|\Xi_{h,l} - \hat{\Xi}_{h,l}\|_2 \overset{(f)}{\geq} \gamma_i(\Xi_{h,l}) - \frac{c}{\sqrt{K}} \overset{(g)}{\geq} \frac{\gamma_i(\Sigma_h)}{D^*} - \frac{c}{\sqrt{K}} \overset{(h)}{\geq} c\frac{\gamma_i(\Sigma_h)}{D^*},$$

where inequality (e) is from the Weyl's inequality (see e.g., Chapter 8 in (Wainwright, 2019)), inequality (f) is from (ii) of event $\mathcal{G}$ in Lemma 18, inequality (g) is from the Assumption 5 (which implies $\gamma_i(\Sigma_h) \leq D^*\gamma_i(\Xi_{h,l})$ from the Weyl's inequality), and inequality (h) is due to $\gamma_i(\Sigma_h) > 0, \forall i \in [\text{Rank}_h]$ and the sufficiently large $K$.

With Assumption 6, we can similarly obtain

$$\sum_{h\in[H]} \mathbb{E}_{\pi^*,\mathcal{M}}\left[ \sqrt{\frac{dH^2}{L^2}}\|\phi(s_h,a_h)\|_{\Upsilon_h^{-1}} + \sqrt{\frac{dH^2}{L}} \right]$$

$$\leq \sqrt{\frac{dH^2}{L^2}} \sum_{h\in[H]} \sqrt{\sum_{l\in[L]}\sum_{i\in[\text{Rank}_l]} \frac{\gamma_i(\Sigma_l)/K}{1/(KL) + \gamma_i(\hat{\Xi}_{h,l})}} + \sqrt{\frac{dH^4}{L}}$$

$$\leq c\sqrt{\frac{dH^2}{L^2}} \sum_{h\in[H]} \sqrt{d(L - L^\dagger)L + \frac{D^*dL}{K}} + \sqrt{\frac{dH^4}{L}}$$

$$\leq c\sqrt{\frac{d^2H^4D^*}{LK}} + \sqrt{\frac{d(L + 1 - L^\dagger)H^4}{L}}.$$

$\square$

# G  SUPPORTING LEMMAS

# H  SUPPORTING LEMMAS

## H.1  SUBOPTIMALITY DECOMPOSITION

The suboptimality gap between an output policy $\hat{\pi}$ from an offline RL algorithm and the optimal policy $\pi^*$ can be decomposed as follows.

**Lemma 23** (Lemma 3.1 in Jin et al. (2021b)). *Let $\hat{\pi} = \{\hat{\pi}_h\}_{h\in[H]}$ be the policy such that $\hat{V}_h(s) = \langle \hat{Q}_h(s,\cdot), \hat{\pi}(\cdot|s)\rangle$. For any $\hat{\pi}$ and $s \in \mathcal{S}$ it holds that*

$$Gap(\hat{\pi};\mathcal{M}) = - \sum_{h\in[H]} \mathbb{E}_{\hat{\pi},\mathcal{M}}\left[\zeta_h(s_h,a_h)|s_1 = s\right] + \sum_{h\in[H]} \mathbb{E}_{\pi^*,\mathcal{M}}\left[\zeta_h(s_h,a_h)|s_1 = s\right]$$

$$+ \sum_{h\in[H]} \mathbb{E}_{\pi^*,\mathcal{M}}\left[\left\langle \hat{Q}_h(s_h,\cdot), \pi^*(\cdot|s_h) - \hat{\pi}_h(\cdot|s_h)\right\rangle |s_1 = s\right],$$

where the expectations $\mathbb{E}_{\hat{\pi},\mathcal{M}}$ and $\mathbb{E}_{\pi^*,\mathcal{M}}$ are with respect to the trajectories induced by $\hat{\pi}$ and $\pi^*$, and

$$\zeta_h(s,a) := (\mathbb{B}_h \hat{V}_{h+1})(s,a) - \hat{Q}_h(s,a)$$

is the model evaluation error at $(s,a,h) \in \mathcal{S} \times \mathcal{A} \times [H]$.

## H.2 ENHANCED EMPIRICAL BERNSTEIN INEQUALITY

In following, an enhanced version of empirical Bernstein inequality is derived. First, the famous Bernstein inequality is presented, which serves as a starting point of the later generalization.

**Lemma 24** (Bernstein Inequality). *Let $Z_1, Z_2, \cdots, Z_n$ be independent random variables with values $|Z_i - \mathbb{E}[Z_i]| \le c_0$. Then, it holds that*

$$\mathbf{P}\left(\left|\sum_{i=1}^n Z_i - \sum_{i=1}^n \mathbb{E}[Z_i]\right| \ge t\right) \le 2\exp\left(-\frac{t^2}{2\sum_{i=1}^n \mathbb{V}(Z_i) + \frac{2}{3}c_0 t}\right),$$

*where $\mathbb{V}(Z_i)$ denotes the variance of random variable $Z_i$.*

It is noted that the true variance is required in the original form of Bernstein inequality, which is hard to be obtained in practice. Thus, Maurer & Pontil (2009) established an empirical version of Bernstein inequality, where the estimated variance is used instead of the true variance.

**Lemma 25** (Empirical Bernstein Inequality; Theorem 4 in Maurer & Pontil (2009)). *Let $Z, Z_1, Z_2, \cdots, Z_n$ be i.i.d. random variables and let $\delta > 0$ with values $|Z_i - \mathbb{E}[Z]| \le c_0$. Then, with probability at least $1 - \delta$, it holds that*

$$\left|\mathbb{E}[Z] - \frac{1}{n}\sum_{i=1}^n Z_i\right| \le \sqrt{\frac{2\hat{\mathbb{V}}(Z)\ln(4/\delta)}{n}} + \frac{7c_0\ln(4/\delta)}{3n},$$

*where $\hat{\mathbb{V}}(Z)$ denotes the empirical variance of $Z$ with samples $Z_1, \cdots, Z_n$*

However, in Lemma 25, the samples $Z_1, \cdots, Z_n$ are i.i.d, i.e., from the same source, which do not meet the need of handling data from heterogeneous sources in this work. Thus, a further enhanced version is provided in the following.

**Lemma 26** (Enhanced Empirical Bernstein Inequality). *Let $Z_l, Z_{1,l}, Z_{2,l}, \cdots, Z_{n_l,l}$ be i.i.d. random variables with values $Z_{i,l} \in [0,1]$ respectively for each $l \in [L]$, and let $\delta > 0$. Then, denoting*

$$\eta := \frac{1}{L}\sum_{l\in[L]}\mathbb{E}[Z_l] - \frac{1}{L}\sum_{l\in[L]}\frac{1}{n_l}\sum_{i=1}^{n_l} Z_{i,l},$$

*with probability at least $1 - \delta$, it holds that*

$$|\eta| \le \sqrt{\sum_{l\in[L]}\frac{8\hat{\mathbb{V}}(Z_l)\log(\frac{2(L+1)}{\delta})}{L^2 n_l}} + \sqrt{\sum_{l\in[L]}\frac{36\left(\log(\frac{2(L+1)}{\delta})\right)^2}{L^2(n_l)^2}},$$

*where $\hat{\mathbb{V}}(Z_l)$ denotes the empirical variance of $Z_l$ with samples $Z_{1,l}, Z_{2,l}, \cdots, Z_{n_l,l}$*

*Proof.* With $n_{\min} = \min_{l\in[L]}\{n_l\}$ and

$$t := \sqrt{\sum_{l\in[L]}\frac{4\mathbb{V}(Z_l)\log(\frac{2(L+1)}{\delta})}{L^2 n_l}} + \frac{4\log(\frac{2(L+1)}{\delta})}{3Ln_{\min}},$$

which implies that

$$\frac{t^2}{2\sum_{l\in[L]}\sum_{i=1}^{n_l}\mathbb{V}(Z_i^l)/(L^2 n_l^2) + 2t/(3Ln_{\min})}$$

$$\leq \frac{t^2}{\max\{4\sum_{l\in[L]}\mathbb{V}(Z_l)/(L^2 n_l), 4t/(3Ln_{\min})\}}$$

$$\leq \log\left(\frac{2(L+1)}{\delta}\right),$$

from Lemma 24, it holds that

$$\mathbf{P}\left(|\eta| \geq t\right) \leq 2\exp\left(-\frac{t^2}{2\sum_{l\in[L]}\sum_{i=1}^{n_l}\mathbb{V}(Z_i^l)/(L^2 n_l^2) + 2t/(3Ln_{\min})}\right)$$

$$\leq \frac{\delta}{L+1}.$$

Furthermore, with Lemma 27 and a union bound, with probability at least $1 - \frac{L\delta}{L+1}$, it holds that

$$\sqrt{\mathbb{V}(Z_l)} \leq \sqrt{\hat{\mathbb{V}}(Z_l)} + \sqrt{\frac{2\ln(\frac{L+1}{\delta})}{n_l}}, \qquad \forall l \in [L].$$

If the above event happens, it holds that

$$t = \sqrt{\sum_{l\in[L]}\frac{4\mathbb{V}(Z_l)\log(\frac{2(L+1)}{\delta})}{L^2 n_l}} + \frac{4\log(\frac{2(L+1)}{\delta})}{3Ln_{\min}}$$

$$\leq \sqrt{\sum_{l\in[L]}\frac{4\left(\sqrt{\hat{\mathbb{V}}(Z_l)} + \sqrt{\frac{2\log(\frac{L+1}{\delta})}{L^2 n_l}}\right)^2\log(\frac{2(L+1)}{\delta})}{n_l}} + \frac{4\log(\frac{2(L+1)}{\delta}}{3Ln_{\min}}$$

$$\leq \sqrt{\sum_{l\in[L]}\frac{8\hat{\mathbb{V}}(Z_l)\log(\frac{2(L+1)}{\delta})}{L^2 n_l}} + \sqrt{\sum_{l\in[L]}\frac{16\left(\log(\frac{2(L+1)}{\delta})\right)^2}{L^2(n_l)^2}} + \frac{4\log(\frac{2(L+1)}{\delta})}{3Ln_{\min}}$$

$$\leq \sqrt{\sum_{l\in[L]}\frac{8\hat{\mathbb{V}}(Z_l)\log(\frac{2(L+1)}{\delta})}{L^2 n_l}} + \sqrt{\sum_{l\in[L]}\frac{32\left(\log(\frac{2(L+1)}{\delta})\right)^2}{L^2(n_l)^2}} + \frac{32\left(\log(\frac{2(L+1)}{\delta})\right)^2}{9L^2 n_{\min}^2}$$

$$\leq \sqrt{\sum_{l\in[L]}\frac{8\hat{\mathbb{V}}(Z_l)\log(\frac{2(L+1)}{\delta})}{L^2 n_l}} + \sqrt{\sum_{l\in[L]}\frac{36\left(\log(\frac{2(L+1)}{\delta})\right)^2}{L^2(n_l)^2}}.$$

Combining this the bound of $t$ with the concentration inequality proved above, the lemma is proved. $\qquad\square$

**Lemma 27** (Theorem 10 in (Maurer & Pontil, 2009)). *Let $Z, Z_1, \cdots, Z_n$ be i.i.d. random variables with values in $[0, 1]$, and let $\delta > 0$, with probability at least $1 - \delta$, it holds that*

$$\sqrt{\mathbb{V}(Z)} < \sqrt{\hat{\mathbb{V}}(Z)} + \sqrt{\frac{2\ln(1/\delta)}{n}}.$$

### H.3 VARIANCE OF MIXTURE MODELS

**Lemma 28.** *With random variable $X_l$ following distribution $\mathbb{P}_l$ for each $l \in [L]$, the random variable $X$ is assumed to follow the distribution mixture, especially, $\mathbb{P} = \sum_{l\in[L]}\alpha_l\mathbb{P}_l$, where $\alpha = [\alpha_1, \cdots, \alpha_L] \in \Delta_L$, the followings holds*

$$C\mathbb{V}(X) \geq \sum_{l\in[L]}\alpha_l^2\mathbb{V}(X_l),$$

*if*

$$C \geq \frac{\sum_{l\in[L]}\alpha_l^2\mathbb{V}(X_l)}{\sum_{l\in[L]}\alpha_l\mathbb{V}(X_l)}.$$

*Proof.* For each side of the desired inequality, we have

$$\text{LHS} := C\mathbb{V}(X) = C\left(\sum_{l\in[L]}\alpha_l\mathbb{E}[X_l^2] - \left(\sum_{l\in[L]}\alpha_l\mathbb{E}[X_l]\right)^2\right)$$

$$\text{RHS} := \sum_{l\in[L]}\alpha_l^2\mathbb{V}(X_l) = \sum_{l\in[L]}\alpha_l^2\mathbb{E}[X_l^2] - \sum_{l\in[L]}\alpha_l^2[\mathbb{E}X_l]^2.$$

With the above expressions, we can further obtain that

$$\begin{aligned}
\text{LHS} - \text{RHS} &= \sum_{l\in[L]}\alpha_l(C-\alpha_l)\mathbb{E}[X_l^2] - C\left(\sum_{l\in[L]}\alpha_l\mathbb{E}[X_l]\right)^2 + \sum_{l\in[L]}\alpha_l^2[\mathbb{E}X_l]^2 \\
&= \sum_{l\in[L]}\alpha_l(C-\alpha_l)\mathbb{E}[X_l^2] + \sum_{l\in[L]}\alpha_l^2(1-C)[\mathbb{E}X_l]^2 - C\sum_{l\neq n}\alpha_l\alpha_n\mathbb{E}[X_l]\mathbb{E}[X_n] \\
&\geq \sum_{l\in[L]}\alpha_l(C-\alpha_l)\mathbb{E}[X_l^2] + \sum_{l\in[L]}\alpha_l^2(1-C)[\mathbb{E}X_l]^2 - C\sum_{l\neq n}\alpha_l\alpha_n\frac{[\mathbb{E}X_l]^2 + [\mathbb{E}X_n]^2}{2} \\
&= \sum_{l\in[L]}\alpha_l(C-\alpha_l)\mathbb{E}[X_l^2] + \sum_{l\in[L]}\alpha_l^2(1-C)[\mathbb{E}X_l]^2 - C\sum_{l\in[L]}\alpha_l(1-\alpha_l)[\mathbb{E}X_l]^2 \\
&= \sum_{l\in[L]}\alpha_l(C-\alpha_l)\mathbb{E}[X_l^2] + \sum_{l\in[L]}\alpha_l(\alpha_l-C)[\mathbb{E}X_l]^2 \\
&= \sum_{l\in[L]}\alpha_l(C-\alpha_l)\mathbb{V}(X_l),
\end{aligned}$$

which leads to the lemma. $\square$

## H.4 MISCELLANEOUS

**Lemma 29** (Covering Number of a Euclidean Ball; Lemma D.5 of Jin et al. (2020)). *There exists a set $\mathcal{C}_\varepsilon \in \mathbb{R}^d$ with $|\mathcal{C}_\varepsilon| \leq (1+2R/\varepsilon)^d$ such that for all $x \in \mathbb{S}^{d-1} := \{x \in \mathbb{R}^d : \|x\|_2 \leq R\}$ there exists a $y \in \mathcal{C}_\varepsilon$ with $\|x-y\|_2 \leq \varepsilon$.*

**Lemma 30** (Lemma H.4 of Min et al. (2021)). *Let $\psi : \mathcal{S} \times \mathcal{A} \to \mathbb{R}^d$ satisfying $\|\psi(s,a)\|_2 \leq C$ for all $(s,a) \in \mathcal{S} \times \mathcal{A}$. For any $T > 0$ and $\kappa > 0$, define $\bar{\mathbf{G}}_T = \sum_{\tau\in[T]}\psi(s^\tau,a^\tau)\psi(s^\tau,a^\tau)^\top + \kappa I_d$ where $(s^\tau, a^\tau)$'s are i.i.d. samples from some distribution $\nu$ over $\mathcal{S} \times \mathcal{A}$. Then, for any $\delta \in (0,1)$, with probability at least $1 - \delta$, it holds that*

$$\left\|\frac{\bar{\mathbf{G}}_T}{T} - \mathbb{E}_\nu\left[\frac{\bar{\mathbf{G}}_T}{T}\right]\right\|_2 \leq \frac{4\sqrt{2}C^2}{\sqrt{T}}\left(\log\frac{2d}{\delta}\right)^{1/2}.$$

