# OpenReview forum: "Offline RL of the Underlying MDP from Heterogeneous Data Sources"
_ICLR.cc/2023/Conference — Submitted to ICLR 2023_

### Official Review · Reviewer_HaHm · 2022-10-23

**Confidence:** 4
**Correctness:** 2
**Technical Novelty And Significance:** 2
**Empirical Novelty And Significance:** Not applicable
**Recommendation:** 3

**Clarity, Quality, Novelty And Reproducibility:**

As stated above, the paper problem setting of the paper is not clear enough for the reviewer to evaluate the strength of the results. With a rough guess of the problem setting, the reviewer believe that the result does not bring any insight. The design of the pessimistic algorithm is not novel given previous works in offline RL.

**Strength And Weaknesses:**

Strength: The paper studies the offline heterogeneous data sources setting, which is more practical than previous works.
Weakness: The statement of the problem setting (learning target) is not very clear. The theoretical result seems a direct application of previous approaches on offline RL for tabular/linear MDPs. In particular, I believe the definition of underlying MDP in Definition 1 is rather confusing. From the statement, it is not clear what the reward and transition kernel of such an MDP are. Are they the reward and transition kernels of a random $ \mathcal{M}_l$? From Definition 1, it seems that the answer is yes and the sampled $l$ can be different across different time step $h$ according to the i.i.d. statement. If this is true, it is hard to understand the real meaning of the learning target defined in Section 2.3 since such an MDP is not what will happen in the real world, but a somewhat mixture model only existing in imagination.



**Summary Of The Paper:**

This paper studies offline reinforcement learning with heterogeneous data sources. The paper propose a new algorithm applying pessimism to handle the randomness from both the sample and the source. It proves the sample efficiency of such an algorithm. The theoretical analysis is based on tabular and linear MDPs.

**Summary Of The Review:**

Given the weakness of the paper stated above, the reviewer believes that the paper is not ready to be accepted.

---

> ### Author Response · Authors · 2022-11-18
> **Responses for Reviewer HaHm (Part I)**
>
> We thank the reviewer for the comments and suggestions. Hopefully, the problem formulation and theoretical contributions can be clarified with the following responses. We will be more than happy to answer any further questions you may have.
>
> **Q1**: The statement of the problem setting (learning target) is not very clear. In particular, I believe the definition of underlying MDP in Definition 1 is rather confusing. From the statement, it is not clear what the reward and transition kernel of such an MDP are. Are they the reward and transition kernels of a random $\mathcal{M}_l$ From Definition 1, it seems that the answer is yes and the sampled $l$ can be different across different time step $h$ according to the i.i.d. statement. If this is true, it is hard to understand the real meaning of the learning target defined in Section 2.3 since such an MDP is not what will happen in the real world, but a somewhat mixture model only existing in imagination.
>
> **A1**: We would like to provide a few clarifications regarding the problem formulation.
>
> - We emphasize that the learning target is the MDP $\mathcal{M} = (H, \mathcal{S}, \mathcal{A}, P, r)$ and the learned policy will be used directly on $\mathcal{M}$. The complication is that learning such a policy must rely on offline data that may be collected from perturbed versions of $\mathcal{M}$.
>
> - To model this scenario, we consider $L$ available datasets and each of them is sampled from a data source MDP $\mathcal{M}_l = (H, \mathcal{S}, \mathcal{A}, P_l, r_l)$, which may have different transition kernels and rewards than the target MDP $\mathcal{M}$. This consideration is highly practical as motivated in Section 1 (e.g., offline dialogues are typically collected from different people with varying language habits), and also well-adopted in offline meta-RL studies.
>
> - While being heterogeneous, this work considers source MDPs are still related to the target MDP. Briefly, source MDPs $\\{\mathcal{M}_l: l\in [L]\\}$ are randomly perturbed versions of the target MDP $\mathcal{M}$. More rigorously, we quote Assumption 1 (original Definition 1) as follows:
>     > **Assumption 1** (Task--source Relationship).
>     Data source MDPs $\\{\mathcal M_l: l\in [L]\\}$ are independently sampled from an unknown distribution $g$ such that for each $(l, s,a,h)\in [L] \times \mathcal{S}\times \mathcal{A}\times [H]$, the reward $r_{h,l}(s,a)$ is a random variable with mean $r_{h}(s,a)$, and the transition vector $P_{h,l}(\cdot|s,a)$ is a random vector with mean $P_{h}(\cdot|s,a)$. In particular, random variables and random vectors $\\{r_{h,l}(s,a), P_{h,l}(\cdot|s,a): (s, a, h, l)\in \mathcal{S}\times \mathcal{A}\times [H] \times [L]\\}$ are independent with each other.
>
>     One of our main contributions is we showed it is possible to efficiently learn the target MDP $\mathcal{M}$ using *finite* data samples from *finite* randomly perturbed data source MDPs $\\{\mathcal{M}_l : l\in [L]\\}$.
>
> - To avoid further confusion, we have made several major revisions:
>     1. The terminology, "underlying MDP", is now replaced in the paper with a more clear one, "target MDP".
>     2. The title is revised as "Offline Reinforcement Learning from Randomly Perturbed Data Sources" to highlight problem setting: this work targets learning with datasets sampled from multiple perturbed versions of the target MDP.
>     3. The target MDP is stated first in Section 2.1 as the targeted task of the agent's interest. The available datasets are then specified to highlight that they are sampled from multiple different data sources. With the target MDP and datasets introduced, their relationship is formulated in Section 2.2 with Assumption1 quoted above.

---

> > ### Author Response · Authors · 2022-11-18
> > **Responses for Reviewer HaHm (Part II)**
> >
> > **Q2**: The theoretical result seems a direct application of previous approaches on offline RL for tabular/linear MDPs.
> >
> > **A2**: In the following, we highlight the theoretical novelties of this work.
> > - The major challenge in this work is that there are **two types of uncertainties** to be considered: **sample uncertainties** from the finite number of data samples per data source, and **source uncertainties** due to a finite number of data sources. A brief discussion is provided in the following while more details can be found in the paragraph "Technical Challenges" in Sec. 3.
> >     - For the sample uncertainties, previous offline RL works only handle one dataset, while this work needs to aggregate information from multiple datasets. Thus, the proposed algorithms need to jointly (instead of individually) consider sample uncertainties from each dataset. Concretely, the penalty term $\Gamma^{\alpha}_h(s,a)$ on page 5 for HetPEVI does not treat each source individually by direct summing up their sample uncertainties. Instead, it is carefully crafted to measure the joint sample uncertainties from all sources.
> >     - The latter source uncertainties is unique in this work. In particular, previous offline RL studies always assume datasets directly sampled from the target MDP. However, the available datasets in this work are from randomly perturbed versions of the target MDP. Thus, even with perfect knowledge of each data source, the target MDP may not be fully revealed, and it is essential to consider the source uncertainty.
> >
> > - Moreover, this work explicitly characterizes the costs and benefits of learning with randomly perturbed data sources. Especially, **the lower bound in Theorem 1 derived during the revision** indicates that an additional unavoidable performance loss occurs due to the indirect access to the target MDP, which can only be reduced by increasing the diversity of data sources, i.e., larger $L$. On the other hand, with multiple available datasets, it is sufficient for them to collectively (instead of individually) provide a good data coverage, which complements previous discussions on coverage requirements in offline RL.

---

### Official Review · Reviewer_LDfB · 2022-10-24

**Confidence:** 4
**Correctness:** 4
**Technical Novelty And Significance:** 2
**Empirical Novelty And Significance:** Not applicable
**Recommendation:** 5

**Clarity, Quality, Novelty And Reproducibility:**

The paper is relatively clear and the results are rigorous and sound. The novelty and technical contributions of the paper are limited.

**Strength And Weaknesses:**

**Strengths:**
- The setting considered in this paper is reasonable (used in prior works such as bandit literature and Mitchell et al. 2021) and relevant to practice. Indeed, in practical settings, the offline dataset is likely to be collected from various sources.
- Rigorous theoretical study of the heterogeneous setting is presented. The main technical contribution is an aggregation of sample uncertainties and source uncertainties.
- Theoretical analysis provides an interesting insight into data source diversity: based on the performance upper bound, collecting samples from more data sources appears to be more helpful than collecting more data from a single source.

Weaknesses:
- Despite the fact that the exact setting considered in this work has not been studied in the past, the technical contributions and novelty are limited. In particular, the algorithm is a straightforward extension of pessimistic value iteration (PEVI) to the setting considered in this paper. Moreover, techniques such as reference-advantage decomposition and Bernstein-type penalties have been widely used in prior work.

**Questions/Comments:**
- What happens if the data sources are hidden (e.g. it is unknown whether a trajectory comes from a particular source)?
- Providing information-theoretic lower bounds and comparing with upper bound on HetPEVI-Adv in this setting strengthens the paper.
- It seems better to mention the paper's technical contributions on top of the Bernstein penalty and reference-advantage decomposition (such as aggregation) instead of these two techniques in the abstract.

**Summary Of The Paper:**

The paper theoretically studies offline RL in the setting where the offline dataset is collected from multiple related but heterogeneous environments. The authors start by studying tabular setting. The paper presents the HetPEVI algorithm that combines the pessimistic value iteration (PEVI) algorithm with penalty terms that involves an aggregation of uncertainties that stem from both source and sample uncertainties. A finite-sample upper bound is proved for the HetPEVI algorithm. The authors then analyze a variant of the algorithm called HetPEVI-Adv that uses the Bernstein penalty and reference-advantage decomposition technique of Zhang et al 2020. Lastly, the authors extend the algorithm to the linear MDP setting and prove a performance upper bound for it.

**Summary Of The Review:**

Despite the rigor and soundness of the problem statement, algorithmic design, and theoretical results, the insights and technical contributions of this work are limited.

---

> ### Author Response · Authors · 2022-11-18
> **Responses for Reviewer LDfB**
>
> We sincerely appreciate the reviewer's efforts and helpful comments. In the following, some responses are provided, which hopefully can clarify your concerns. It will be our pleasure to answer any further questions that you may have.
>
> **Q1**: What happens if the data sources are hidden (e.g. it is unknown whether a trajectory comes from a particular source)?
>
> **A1**: We first want to note that in practice, it is often reasonable to separate data collected from different environments. In the chatbot example used in the paper, one can imagine one dataset is collected from adults and another from children.
>
> The scenario with unknown trajectory identities proposed by the reviewer is a really interesting setting, but considerably harder than the studied one. Some preliminary thoughts are provided in the following.
>
> - Without the identity information, a straightforward idea is to directly count the overall visitations and perform estimations. In this way, the estimation of $P_h(s'|s,a)$ would be $\frac{\sum_{l\in [L]}N_{h,l}(s,a, s')}{\sum_{l\in [L]}N_{h,l}(s,a)}$, where $N_{h,l}(s,a)$ and $N_{h,l}(s,a)$ are the number of visitation on $(s,a)$ and $(s,a,s')$ in the $l$-th datasets. However, it is hard to ensure the quality of such estimations. Instead, with the identity information, estimations can be properly performed, e.g., $\frac{1}{L}\sum_{l\in [L]}\frac{N_{h,l}(s,a, s')}{N_{h,l}(s,a)}$ for $P_h(s'|s,a)$.
>
> - Another more reasonable solution is to cluster the trajectories and then perform the algorithms proposed in this work. On one hand, for practical implementations, any trajectory clustering algorithm would be sufficient. On the other hand, it is theoretically challenging to ensure the correctness of such clusters. To the best of our knowledge, one candidate clustering method is proposed in [R1] for the study of latent MDP; however, it requires strong prior knowledge about the rewards and transitions of each source MDP.
>
> We have added this direction as an open question in Appendix A.2. It is our hope that this work can inspire further investigations on offline RL with multiple data sources.
>
> **Q2**: Providing information-theoretic lower bounds and comparing with upper bound on HetPEVI-Adv in this setting strengthens the paper.
>
> **A2:** As suggested by the reviewer, **a novel information-theoretical lower bound is proved in Theorem 1 during the revision**. This lower bound characterizes the performance loss due to finite data samples and finite data sources, respectively.
>
> Comparing the performance upper bounds of HetPEVI (Theorem 2) and HetPEVI-Adv (Theorem 6 in Appendix E) with the lower bound, it can be observed that they are optimal in the dependency on parameters $C^*$, $S$, $L$, and $K$. Regarding the $H$ dependency, there exists additional $\sqrt{H}$  and $H$ factors in the performance loss from finite samples and finite sources, respectively. The former can be addressed by HetPEVI-Adv with the Bernstein-type penalty design; however, the latter is left open for future investigations. These discussions are added to Section 4.1 (after Theorem 2) during the revision.
>
> **Q3**: It seems better to mention the paper's technical contributions on top of the Bernstein penalty and reference-advantage decomposition (such as aggregation) instead of these two techniques in the abstract.
>
> **A3**: We thank the reviewer for this constructive comment. During the revision, we removed the discussions of the Bernstein penalty and reference-advantage decomposition from the abstract. Instead, as suggested, the technical contributions are highlighted, espcially about considering two types of uncertainties (i.e., sample uncertainties and sources uncertainties) and aggregating information from all sources. Moreover, inspired by Reviewer hDmb, we have also provided additional highlights on the costs (i.e., additional performance loss) and benefits (i.e., collective coverage) of learning with multiple randomly perturbed datasets in this abstract.
>
>
> [R1] Kwon, Jeongyeol, et al. "RL for latent MDPs: Regret guarantees and a lower bound." Advances in Neural Information Processing Systems 34 (2021): 24523-24534.

---

### Official Review · Reviewer_cUra · 2022-10-24

**Confidence:** 3
**Correctness:** 4
**Technical Novelty And Significance:** 3
**Empirical Novelty And Significance:** Not applicable
**Recommendation:** 6

**Clarity, Quality, Novelty And Reproducibility:**

-

**Strength And Weaknesses:**

I am curious on how these theoretical results compare to other offline RL theory that lower-bounds the  on the Value with lipschitz continuity assumptions over the transition and reward dynamics, i.e. the DeepAveragers framework[1]. As the Averagers framework already treats the seen rewards as samples of the underlying MDP, the Lipschitz constant over the formulation with heterogeneous datasets will simply be the max(Constant from Sample Uncertainity, Constant from source uncertainty). This also points out to the underlying notion that given the Constant for Sample uncertainty may be higher and hence it may be better to have diverse data sources over larger data samples. I would love to know your thoughts on how and if these two lower bounding approaches connect with each other.
Moreover, I understand that the limited data sources incur an additional performance loss that cannot be reduced by increasing the amount of data samples. However, It would be nice to know if there was some elaboration on when is it very important to have data from multiple sources. For example I can imagine that if the variance between sources is zero, the data samples from a new source or an old source should count the same.

 [1] Shrestha, A., Lee, S., Tadepalli, P., & Fern, A. (2021). DeepAveragers: Offline Reinforcement Learning by Solving Derived Non-Parametric MDPs. ArXiv, abs/2010.08891.

**Summary Of The Paper:**

This paper sheds some theoretical light on the problem offline RL with data sets from heterogeneous sources; which closely relates to the field of meta- Reinforcement Learning. Here they pose this problem as learning an underlying Markov Decision Process (MDP) M from samples of multiple randomly perturbed variation of M. To this end the authors propose, HetPEVI as a method to jointly consider the uncertainties that arise from the limited data samples as well as the limited number of perturbed MDPs that we sampled the data from. The authors build on HetPEVI to propose HetPEVI-Adv which theoretically allows for less conservative estimates by incorporating priors for sample independence and shared variance information. Finally, HetPEVI-Lin is introduces as a specialized adaptation of HetPEVI for linear MDPs that builds on the linear formulations of original PEVI and uncertainity quantification of HetPEVI.

**Summary Of The Review:**

Overall the paper nicely extends the original work of PEVI to the setting of heterogeneous datasets in a principled manner. I am leaning towards accepting the paper as it provides principled theoretical foundations for offline RL with heterogeneous sources albeit having some empirical results for some real world problems would bolster the paper greatly.

---

> ### Author Response · Authors · 2022-11-18
> **Responses for Reviewer cUra**
>
> We thank the reviewer for the constructive comments and look forward to further discussions with the reviewer.
>
> **Q1**: I am curious on how these theoretical results compare to other offline RL theory that lower-bounds the  on the Value with lipschitz continuity assumptions over the transition and reward dynamics, i.e. the DeepAveragers framework [R1]. As the Averagers framework already treats the seen rewards as samples of the underlying MDP, the Lipschitz constant over the formulation with heterogeneous datasets will simply be the max(constant from sample uncertainity, constant from source uncertainty). This also points out to the underlying notion that given the constant for sample uncertainty may be higher and hence it may be better to have diverse data sources over larger data samples. I would love to know your thoughts on how and if these two lower bounding approaches connect with each other.
>
> **A1**: We thank the reviewer for bringing this interesting connection to our attention.  Some discussions are provided in the following.
>
> In our mind, this work can be philosophically connected to [R1] as:
> - For a state-action pair, this work considers that multiple datasets (from different sources) contain information about it, while [R1] looks for similar state-action pairs with small distances in the dataset; thus, they can be related by thinking of available datasets in this work as the available neighbors in [R1].
> - Then, the Lipschitz continuity assumption in [R1] serves a similar role as Assumption 1 in this work (originally Definition 1) to establish the connection between desired task information with the available datasets. From this perspective, the first term in Theorem 3.1 [R1] can be interpreted as coming from the source uncertainty while the second term is from the sample uncertainty.
>
> However, we also note that there are still differences between the Lipschitz assumption in [R1] and the stochastic relationship (Assumption 1) in this work. Especially, the former is a worst-case consideration that would not characterize the concentration of involving more data sources. More detailed discussions have been added to Appendix A.1 of relative works.
>
>
> **Q2**:
> Moreover, I understand that the limited data sources incur an additional performance loss that cannot be reduced by increasing the amount of data samples. However, It would be nice to know if there was some elaboration on when is it very important to have data from multiple sources. For example, I can imagine that if the variance between sources is zero, the data samples from a new source or an old source should count the same.
>
>
> **A2**: We thank the reviewer for this helpful suggestion. The current formulation (Assumption 1) does not pose any restriction on the relationship between source MDPs (except their independent generation). Thus, the designed algorithms and derived bounds are intended for the worst case, i.e., accommodating an arbitrary unknown variance between sources. If there is additional information about the variance between sources, it is feasible to incorporate it into the design and obtain variance-dependent results, which would cover the zero-variance case mentioned by the reviewer. In particular, as noted in Remark 2 of Section 3, if the rewards and transition vectors are generated via $\sigma$-sub-Gaussian distributions, the penalty can be designed as $\Gamma^\beta_h(s,a) = c\sqrt{\sigma^2H^2/L}$, which shrinks with a smaller variance $\sigma$.
>
>
> **Reference**
>
> [R1] Shrestha, A., Lee, S., Tadepalli, P., & Fern, A. (2021). DeepAveragers: Offline Reinforcement Learning by Solving Derived Non-Parametric MDPs. ArXiv, abs/2010.08891.

---

### Official Review · Reviewer_hDmb · 2022-10-27

**Confidence:** 3
**Clarity, Quality, Novelty And Reproducibility:** See questions above. The reproducibil…
**Correctness:** 3
**Technical Novelty And Significance:** 2
**Empirical Novelty And Significance:** Not applicable
**Recommendation:** 5

**Strength And Weaknesses:**


$\textbf{Strength:}$

The offline MDP challenge with the heterogeneous data source is important and has potential usage in practice. The algorithms are well-motivated and ready for practical realization. The analyses are well presented, with all the results discussed thoroughly.




$\textbf{Questions and suggestions:}$


The organization of this paper could be improved. I think this paper needs better motivation on the math problem it attempts to address. In particular,

$\textbf{Q1}$
Why do we want to solve a "mean MDP" (the underlying MDP) problem? How does solving such an underlying MDP contribute to the agent's performance in local environments (which the agent is supposed to serve)? Why not consider transitions and rewards with the variation that aligns with data source behavior? An ordinary (offline) RL algorithm should be able to handle such a problem.

$\textbf{Q2}$
In Definition 1, why are $r_{h, \ell}$ (same for $P_{g, \ell}$) identically distributed across (h, \ell)? Does the analysis hinge on such an identical assumption? Given the motivation in the introduction, I think a more natural setting is if $r_{h, \ell}$ are different distributions for different $\ell$ due to the data source preference. They could still share the same mean, though. In addition, Definition 1 casts restrictions on the set of MDPs (iid distribution), so it is not exactly a definition of underlying MDP but a definition of both underlying MDP and the associated MDP set. The authors may clarify such a point for revisions to avoid misunderstanding.


Besides, the extensions with the Bernstein-type technique and linear MDPs are less important and can be put into the appendix.

In addition, I have several inquiries.

$\textbf{Q3}$
The technical contribution seems marginal, given the results in, e.g., [1][2][3] (and the numerous previous Hoeffding- and Bernstein- type analyses of tabular and linear MDPs) and that all the rewards and transitions are iid distributed. Could the authors highlight their analysis's technical challenges and subtle parts, comparing against [1][2][3]?

$\textbf{Q5}$
Is the achieved sample complexity in Theorem 1 optimal? What would be the lower bound for such a type of problem?

$\textbf{Q5}$
(Minor) Assumption 1 seems strong for such a problem. Is it necessary that data collected from each element of the MDP set has to explore the environment sufficiently well? Is it possible that they do not have good coverage individually but together cover the underlying MDP sufficiently well?

[1] Jin et al., Is Pessimism Provably Efficient for Offline RL? (2021)
[2] Rashidinejad et al., Bridging Offline Reinforcement Learning and Imitation Learning: A Tale of Pessimism. (2021)
[3] Ming et al., Near-optimal offline reinforcement learning with linear representation: Leveraging variance information with pessimism. (2022)

**Summary Of The Paper:**

This paper aims to tackle the offline MDP challenge with the heterogeneous data source. To this end, the authors propose an underlying MDP setup where the offline data are sampled from an (iid) variation of the underlying MDP. The authors further present several algorithms and the associated analyses (for both tabular case and linear MDP case) to show that the proposed algorithms are sample efficient.

**Summary Of The Review:**

The problem is interesting but the underlying math model needs better motivation. The presentation could be improved to highlight the motivation and the technical challenges of the paper.

---

> ### Author Response · Authors · 2022-11-18
> **Responses for Reviewer hDmb (Part I)**
>
> We thank the reviewer for the helpful comments and would like to provide the following responses. Hopefully, these discussions can clarify the reviewer's concerns, and we will be more than happy to answer any further questions you may have.
>
> **Q1**: Why do we want to solve a "mean MDP" (the underlying MDP) problem? How does solving such an underlying MDP contribute to the agent's performance in local environments (which the agent is supposed to serve)? Why not consider transitions and rewards with the variation that aligns with data source behavior? An ordinary (offline) RL algorithm should be able to handle such a problem.
>
> **A1**: We hope the following illustrations and corresponding revisions of the paper can clarify your concerns.
>
> - First, we note that the underlying MDP is the agent's learning objective. However, the available datasets are related to but not necessarily aligned with it. Especially, this work considers a novel model where the data sources are randomly perturbed versions of the task.
>     - If data directly from the target task are available, we agree with the reviewer that it is sufficient to use ordinary offline RL. However, it is practically hard to ensure that all data comes directly from the target task. With the example of training a chatbot in Section 1, the task is to learn the language structure; however, offline dialogues are typically collected from different people with varying language habits. Growing empirical interests in this direction have been witnessed under the framework of offline meta-RL and this work targets at complementing them with additional theoretical understandings.
>
> - To avoid further confusion, we have made several major revisions:
>     1. The terminology, "underlying MDP", is now replaced in the paper with a more clear one, "target MDP".
>     2. The title is revised as "Offline Reinforcement Learning from Randomly Perturbed Data Sources" to highlight the major difference with previous offline RL works: this work targets learning with datasets sampled from multiple randomly perturbed versions of the target MDP.
>     3. The problem formulation section is reconstructed. In particular, the target MDP is stated first in Section 2.1 as the agent's targeted task. The available datasets are then specified to highlight that they are sampled from multiple different data sources. With the target MDP and datasets introduced, their relationship is formulated in Section 2.2.
>
> **Q2**: In Definition 1, why are $r_{h,l}$ (same for $P_{h,l}$) identically distributed across (h, \ell)? Does the analysis hinge on such an identical assumption? Given the motivation in the introduction, I think a more natural setting is if $r_{h,l}$ are different distributions for different $l$ due to the data source preference. They could still share the same mean, though. In addition, Definition 1 casts restrictions on the set of MDPs (iid distribution), so it is not exactly a definition of underlying MDP but a definition of both underlying MDP and the associated MDP set. The authors may clarify such a point for revisions to avoid misunderstanding.
>
> **A2**: We sincerely thank the reviewer for the constructive suggestion regarding the definition between the target MDP and the source MDPs. The original Definition 1 is now restated as an assumption for the relationship between them; briefly, the data source MDPs are randomly perturbed versions of the target MDP. To facilitate your reading, Assumption 1 is quoted in the following:
>
> > **Assumption 1** (Task-source Relationship).
>     Data source MDPs $\\{\mathcal M_l : l \in [L] \\}$ are independently sampled from an unknown distribution $g$ such that for each $(l, s,a,h)\in [L] \times \mathcal{S}\times \mathcal{A}\times [H]$, the reward $r_{h,l}(s,a)$ is a random variable with mean $r_{h}(s,a)$, and the transition vector $P_{h,l}(\cdot|s,a)$ is a random vector with mean $P_{h}(\cdot|s,a)$. In particular, random variables and random vectors $\\{r_{h,l}(s,a), P_{h,l}(\cdot|s,a): (s, a, h, l)\in \mathcal{S}\times \mathcal{A}\times [H] \times [L]\\}$ are independent with each other.
>
> Furthermore, regarding the question about the identical distribution, we believe our formulation aligned with the reviewer's suggestion. In particular, for each $l$, the reward $r_{h,l}(s,a)$ and the transition probability $P_{h,l}(\cdot|s,a)$ are randomly perturbed from the target MDP, i.e., $r_{h}(s,a)$ and $P_h(\cdot|s,a)$. As a result, for different $l$ and $m$, their rewards and transitions are not necessarily aligned, i.e., $r_{h,l}(s,a)\neq r_{h, m}(s,a)$ and $P_{h,l}(\cdot|s,a) \neq P_{h,m}(\cdot|s,a)$.
>
> P.S. We also note that here $r_{h,l}(s,a)$ is a deterministic reward for the state-action pair $(s,a)$ at step $h$ on the MDP $\mathcal{M}_l$, instead of a reward distribution.

---

> > ### Author Response · Authors · 2022-11-18
> > **Responses for Reviewer hDmb (Part II)**
> >
> > **Q3**:
> >  The technical contribution seems marginal, given the results in, e.g., [R1, R2, R3] (and the numerous previous Hoeffding- and Bernstein- type analyses of tabular and linear MDPs) and that all the rewards and transitions are iid distributed. Could the authors highlight their analysis's technical challenges and subtle parts, comparing against [R1, R2, R3]?
> >
> >
> > **A3**: We summarize the technical challenges in this work compared with previous pessimistic-style offline RL studies as follows.
> >
> > - **Two types of uncertainties.** The major challenge in this work is that there are two types of uncertainties to be considered: **sample uncertainties** from the finite number of data samples per data source, and **source uncertainties** due to a finite number of data sources. Instead, previous offline RL works, including [R1, R2, R3], only consider the former, i.e., the sample uncertainties. It is essential to incorporate the latter source uncertainties in this work as even with perfect knowledge of each data source, the target MDP may not be fully revealed. More discussions can be found in the paragraph "Technical Challenges" in Sec. 3.
> >
> > - **Aggregation of Sample Uncertainties.** Unique challenges also exist in how to jointly aggregate sample uncertainties from all data sources. In particular, previous investigations, including [R1, R2, R3], only deal with one single data source, rather than multiple ones as in this work. This difference concretely reflects in the design of penalties for both tabular and linear settings. In particular, for the HetPEVI algorithm, the penalty term $\Gamma_h^{\alpha}(s,a)$ is a joint measure of sample uncertainties from all sources; thus it provides a $1/\sqrt{L}$ speed-up in performance guarantees compared with directly summing up sample uncertainties of each data sources. More discussions can be found in the paragraph "Penalties to Aggregate Sample Uncertainties" in Section 3).
> >
> > **Q4**:
> > Is the achieved sample complexity in Theorem 1 optimal? What would be the lower bound for such a type of problem?
> >
> > **A4:**
> > - **A new lower bound analysis is now added to the paper in Theorem 1.** It highlights that the limited number of randomly perturbed data sources introduces an additional unavoidable performance loss, which cannot be reduced by involving more data from each source.
> >
> > - Compared with the lower bound, the performance of HetPEVI (in Theorem 2) is tight regarding its dependency on parameters $C^*$, $S$, $L$, and $K$. There exists additional $\sqrt{H}$  and $H$ factors in the performance loss from finite samples and finite sources, respectively. The former can be addressed by HetPEVI-Adv (in Theorem 6 of Appendix E.2) with the Bernstein-type penalty design; however, the latter is left open for future investigations.
> >
> > **Q5**: Assumption 1 seems strong for such a problem. Is it necessary that data collected from each element of the MDP set has to explore the environment sufficiently well? Is it possible that they do not have good coverage individually but together cover the underlying MDP sufficiently well?
> >
> > **A5**: Thank you for this really constructive suggestion. Indeed, as long as the datasets collectively have a good coverage of the target MDP, efficient learning is achievable. We have made major revisions to highlight this benefit introduced by learning with multiple datasets. Especially, Section 4.2 (in particular, Theorem 3) is added for the tabular case; Theorem 5 is added for the linear case. Please find more details therein.
> >
> >
> > **References**
> >
> > [R1] Jin et al., Is Pessimism Provably Efficient for Offline RL? (2021)
> >
> > [R2] Rashidinejad et al., Bridging Offline Reinforcement Learning and Imitation Learning: A Tale of Pessimism. (2021)
> >
> > [R3] Ming et al., Near-optimal offline reinforcement learning with linear representation: Leveraging variance information with pessimism. (2022)

---

### Author Response · Authors · 2022-11-27
**Post rebuttal**

Dear reviewers:

Thank you for your insightful comments. Point-by-point responses have been provided and the corresponding updates have been made to the paper, which are marked by the highlighted text. If you have additional concerns or feedback after reading our responses, please do not hesitate to let us know, and we will be happy to answer them.

Thanks,

Authors of Paper 3430

---

### Decision · Program_Chairs · 2023-01-20

**Decision:**

Reject

**Justification For Why Not Higher Score:**

The presentation needs much improvement, including the exact problem setting.  The technical contribution and novelty also requires better clarification.  Overall, the paper needs another round of revision and review.

**Justification For Why Not Lower Score:**

N/A

**Metareview: Summary, Strengths And Weaknesses:**

This paper addresses offline reinforcement learning with the heterogeneous data source, where each MDP is considered as sampled iid from and underlying MDP.  The proposed HetPEVI algorithm accounts for both the sample uncertainty per data source and the source uncertainty.  An improvement called HetPEVI-Adv was further proposed that employs Bernstein penalty and reference-advantage decomposition.  Extension to linear function approximation was also presented.

The paper addresses an important setting in offline RL.  The presentation needs much improvement, including the exact problem setting.  The technical contribution and novelty also requires better clarification.  Overall, the paper needs another round of revision and review.